# Variations in the Summer Oceanic $p$CO$_2$ and Carbon Sink in the Prydz Bay Using the SOM Analysis Approach

**Suqing Xu[1], Keyhong Park[2*], Yanmin Wang[3], Liqi Chen[1*], Di Qi[1], Bingrui Li[4]**

1.  Key Laboratory of Global Change and Marine-Atmospheric Chemistry, Third Institute of Oceanography, Ministry of Natural Resources, Xiamen 361005, PR China.
2.  Division of Polar Ocean Sciences, Korea Polar Research Institute, Incheon 21990, South Korea.
3.  Haikou Marine Environment Monitoring Central Station, State Oceanic Administration, Haikou 570100, China.
4.  Polar Research Institute of China, Shanghai 200136, China.

Correspondence to: Liqi Chen (chenliqi@tio.org.cn);

Keyhong Park (keyhongpark@kopri.re.kr)

**Abstract**

This study applies a neural network technique to produce maps of oceanic surface $p$CO$_2$ in the Prydz Bay in the Southern Ocean on a weekly 0.1° longitude · 0.1° latitude grid based on in situ measurements obtained during the 31$^{st}$ CHINARE cruise from February to early March of 2015. This study area was divided into three regions, namely, the Open-ocean region, Sea-ice region and Shelf region. The distribution of oceanic $p$CO$_2$ was mainly affected by physical processes in the Open-ocean region, where mixing and upwelling were the main controls. In the Sea-ice region, oceanic $p$CO$_2$ changed sharply due to the strong change in seasonal ice. In the Shelf region, biological factors were the main control. The weekly oceanic $p$CO$_2$ was estimated using a self-organizing map (SOM) with four proxy parameters (Sea Surface Temperature, Chlorophyll-a concentration, Mixed Layer Depth, and Sea Surface Salinity) to overcome the complex relationship between the biogeochemical and physical conditions in the Prydz Bay region. The reconstructed oceanic $p$CO$_2$ data coincide well with the in situ investigated $p$CO$_2$ data from SOCAT, with a root-mean-square error of 22.14 µatm. The Prydz Bay was mainly a strong CO$_2$ sink in February 2015, with a monthly averaged uptake of 23.57±6.36 TgC. The oceanic CO$_2$ sink is pronounced in the Shelf region due to its lowest oceanic $p$CO$_2$ and peak biological production.

## 1 Introduction

The amount of carbon uptake occurring in the ocean south of 60°S is still uncertain despite its importance in regulating atmospheric carbon and acting as a net sink for anthropogenic carbon (Sweeney et al., 2000, 2002; Morrison et al., 2001; Sabine et al., 2004; Metzl et al., 2006; Takahashi et al., 2012). This uncertainty arises from both the strong seasonal and spatial variations that occur around Antarctica and the difficulty of obtaining field measurements in the region because of its hostile weather and remoteness.

Following the Weddell and Ross seas, the Prydz Bay is the third-largest embayment in the Antarctic continent. Situated in the Indian Ocean section, the Prydz Bay is located close to the Amery Ice Shelf to the southwest and the West Ice Shelf to the northeast, with Cape Darnley to the west and the Zhongshan and Davis stations to the east (Fig. 1). In this region, the water depth increases sharply northward from 200 m to 3000 m.

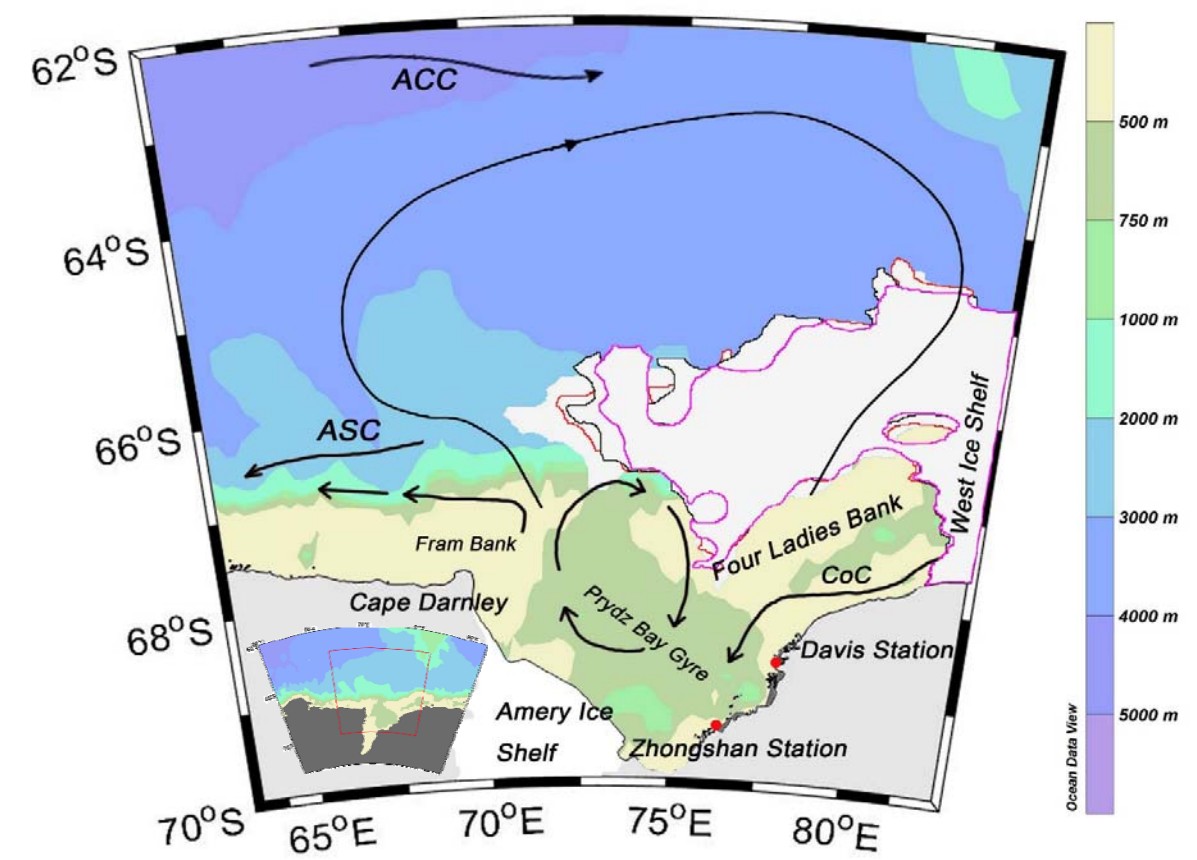

Fig. 1 Ocean circulations in the Prydz Bay derived from Roden et al. (2013), Sun et al. (2013), Wu et al. (2017).

ASC: Antarctic Slope Current; CoC: Antarctic Coastal Current; ACC: Antarctic Circumpolar Current. During

the 4-week cruise, the sea ice extent varied as indicated by the contoured white areas: the pink line is for week-1(20150202-20150209), the black line is for week-2 (20150210-20150217), the red line is for the week-3 (20150218-20150225) and a fourth contoured area is for week-4 (20150226-20150305).

The inner continental shelf is dominated by the Amery Depression, which mostly ranges in depth from 600 to 700 m. This depression is bordered by two shallow banks (<200 m): the Fram Bank and the Four Ladies Bank, which form a spatial barrier for water exchange with the outer oceanic water (Smith and Trégure, 1994). The Antarctic Coastal Current (CoC) flows westward, bringing in cold waters from the east. When the CoC reaches the shallow Fram Bank, it turns north and then partly flows westward, while some of it turns eastward, back to the inner shelf, resulting in the clockwise-rotating Prydz Gyre (see Fig.1). The circulation to the north of the bay is characterized by a large cyclonic gyre, extending from within the bay to the Antarctic Divergence at approximately 63°S (Nunes Vaz and Lennon, 1996; Middleton and Humphries, 1989; Smith et al., 1984; Roden et al., 2013; Wu et al., 2017). The inflow of this large gyre hugs the eastern rim of the bay and favours the onshore intrusions of warmer modified Circumpolar Deep Water across the continental shelf break (Heil et al., 1996). Westward flow along the shelf, which is part of the wind-driven Antarctic Slope Current (ASC), supplies water to the Prydz Bay.

It has been reported that the Prydz Bay is a strong carbon sink, especially in the austral summer (Gibsonab et al.,1999; Gao et al., 2008; Roden et al., 2013). Moreover, studies have shown that the Prydz Bay region is one of the source regions of Antarctic Bottom Water as well as the Weddell and Ross seas (Jacobs and Georgi,1977; Yabukiet al., 2006). It is thus important to study the carbon cycle in the Prydz Bay. However, the analysis of the temporal variability and spatial distribution mechanism of oceanic $p\mathrm{CO_2}$ in the Prydz Bay is limited to cruises or stations due to its unique physical environment and complicated marine ecosystem (Smith et al., 1984; Nunes Vaz et al., 1996; Liu et al., 2003). To estimate regional sea-air $\mathrm{CO_2}$ fluxes, it is necessary to interpolate between in situ measurements to obtain maps of oceanic $p\mathrm{CO_2}$. Such an interpolation approach, however, is still difficult, as observations are too sparse over both time and space to capture the high variability in $p\mathrm{CO_2}$. Satellites do not measure sea surface $p\mathrm{CO_2}$, but they do provide access to the parameters related to the processes that control its variability. The seasonal and geographical variability of surface water $p\mathrm{CO_2}$ is indeed much greater than that of atmospheric $p\mathrm{CO_2}$. Therefore, the direction of sea-air $\mathrm{CO_2}$ transfer is mainly regulated by oceanic $p\mathrm{CO_2}$, and the method of spatially and temporarily interpolating in situ measurements of oceanic $p\mathrm{CO_2}$ has long been used (Takahashi et al., 2002 and 2009; Olsen et al., 2004; Jamet et al., 2007; Chierici et

al., 2009). In earlier studies, a linear regression extrapolation method was applied to expand cruise
data to study the carbon cycle in the Southern Ocean (Rangama et al., 2005; Chen et al., 2011; Xu
et al., 2016). However, this linear regression relied simply on either chlorophyll-a (CHL) or sea
surface temperature (SST) parameters. Thus, this method can not sufficiently represent all
controlling factors. In this study, we applied self-organizing map (SOM) analysis to expand our
observed data sets and estimate the oceanic $p$CO$_2$ in the Prydz Bay from February to early March
of 2015.

The SOM analysis, which is a type of artificial neural network, has been proven to be a useful
method for extracting and classifying features in the geosciences, such as trends in (and between)
input variables (Gibson et al., 2017; Huang et al., 2017b). The SOM uses an unsupervised
learning algorithm (i.e., with no need for a priori, empirical or theoretical descriptions of
input-output relationships), thus enabling us to identify the relationships between the state
variables of the phenomena being analysed, where our understanding of these cannot be fully
described using mathematical equations and thus where applications of knowledge-based models
are limited (Telszewski et al., 2009). In the field of oceanography, SOM has been applied for the
analysis of various properties of seawater, such as sea surface temperature (Iskandar, 2010; Liu et
al., 2006), and chlorophyll concentration (Huang et al., 2017a; Silulwane et al., 2001). In the past
decade, SOM has also been applied to produce basin-scale $p$CO$_2$ maps, mainly in the North
Atlantic and Pacific Ocean, by using different proxy parameters (Lafevre et al., 2005; Friedrich
&Oschlies, 2009a, 2009b; Nakaoka et al., 2013; Telszewski et al., 2009; Hales et al., 2012; Zeng et
al., 2015; Laruelle et al., 2017). SOM has been proven to be useful for expanding the
spatial-temporal coverage of direct measurements or for estimating properties whose satellite
observations are technically limited. One of the main benefits of the neural network method over
more traditional techniques is that it provides more accurate representations of highly variable
systems of interconnected water properties (Nakaoka et al., 2013).

We conducted a survey during the 31$^{st}$ CHINARE cruise in the Prydz Bay (Fig. 2). This study
aimed to apply the SOM method, combined with remotely sensed data, to reduce the
spatiotemporal scarcity of contemporary  $\triangle p$CO$_2$ data and to obtain a better understanding of the
capability of carbon absorption in the Prydz Bay from 63°E to 83°E and 64°S to 70°S from
February to early March of 2015.
The paper is organized as follows. Section 2 provides descriptions of the in situ measurements
and SOM methods. Section 3 presents the analysis and discussion of the results, and section 4
presents a summary of this research.
**2 Data and methods**
**2.1 In situ data**
The in situ underway $p$CO$_2$ values of marine water and the atmosphere were collected during
the 31$^{st}$ CHINARE cruise, when the R/V Xuelong sailed from east to west from the beginning of
February to early March, 2015 (see Fig.2a, b). Sea water at a depth of 5 metres beneath the sea
surface was pumped continuously to the GO system (GO Flowing$p$CO$_2$ system, General Oceanics
Inc., Miami FL, USA), and the partial pressure of the sea surface water was measured by an
infrared analyser (LICOR, USA, Model 7000). The analyser was calibrated every 2.5-3 h using
four standard gases supplied by NOAA's Global Monitoring Division at pressures of 88.82 ppm,
188.36 ppm, 399.47 ppm, and 528.92 ppm. The accuracy of the measured $p$CO$_2$ data is within 2
μatm (Pierrot et al., 2009). Underway atmospheric $p$CO$_2$ data were simultaneously collected by
the GO system. The biological and physical pumps in the ocean (Hardman-Mountford et al., 2009;
Bates et al., 1998a, 1998b; Barbini et al., 2003; Sweeney, 2002), are the key factors controlling the
variation in sea surface $p$CO$_2$.    In terms of the physical pumps, the solubility of CO$_2$ is affected
by temperature and salinity, but the biological pumps, such as, phytoplankton, take up CO$_2$
through photosynthesis while organisms release it through respiration (Chen et al., 2011). There
are several processes that can influence the distribution of oceanic $p$CO$_2$.
Sea ice melt has a significant impact on the local stratification and circulation in polar regions.
During freezing, brine is rejected from ice, thereby increasing the sea surface salinity. When ice
begins to melt, fresher water is added into the ocean, thereby diluting the ocean water, i.e.,
reducing its salinity. Changes in salinity thus record physical processes. In this study, we treat
salinity as an index for changes in sea ice. The underway SST and conductivity data were recorded
by a Conductivity-Temperature-Depth sensor (CTD, Seabird SBE 21) along the cruise track. Later,
sea surface salinity was calculated based on the recorded conductivity and temperature data. The
distributions of underway SST and SSS are shown in Fig.2c and d.
In austral summer, when sea ice started to melt, ice algae were released into the seawater, and
the amount of living biological species and primary productivity increased; thus, high
chlorophyll-a values were observed (Liu et al., 2000; Liu et al., 2003). Previous studies have
reported that the summer sink in the Prydz Bay is biologically driven and that the change in $p$CO$_2$
is often well correlated with the surface chlorophyll-a concentration (Rubin et al., 1998; Gibsonab
et al., 1999; Chen et al., 2011; Xu et al., 2016). The chlorophyll-a value is regarded as an important
controlling factor of $p$CO$_2$. Remote sensing data of chlorophyll-a obtained from MODIS with a
resolution of 4 km (http://oceancolor.gsfc.nasa.gov) were interpolated according to the cruise
track (Fig.2e).
The ocean mixed layer is characterized as having nearly uniform physical properties
throughout the layer, with a gradient in its properties occurring at the bottom of the layer. The
mixed layer links the atmosphere to the deep ocean. Previous studies have emphasized the
importance of accounting for vertical mixing through the mixed layer depth (MLD, Dandonneau,
1995; Lüger et al., 2004). The stability and stratification of this layer prevent the upward mixing of
nutrients and limit biological production, thus affecting the sea-air CO$_2$ exchange. Two main
methods are used to calculate the MLD (Chu and Fan, 2010): one is based on the difference
criterion, and one is based on the gradient criterion. Early studies suggested that the MLD values
determined in the Southern Ocean using the difference criterion are more stable (Brainerd and
Gregg, 1995; Thomson and Fine, 2003). Thus, following Dong et al. (2008), we calculated the
mixed layer depth (see Fig.2f) based on the difference criterion, in which sigma theta changed by
0.03 kg/m$^3$. The MLD values at the stations along the cruise were later gridded linearly to match
the spatial resolution of the underway measurements.

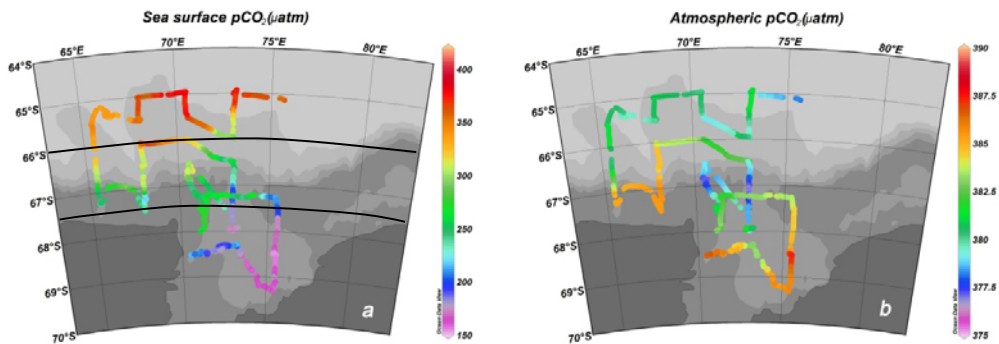


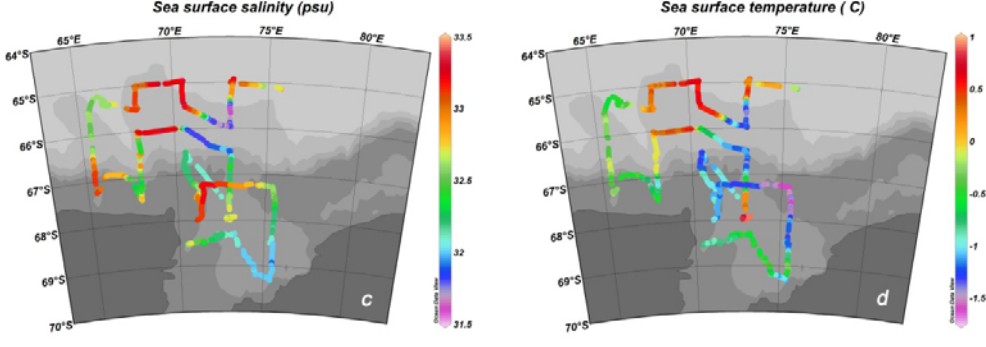

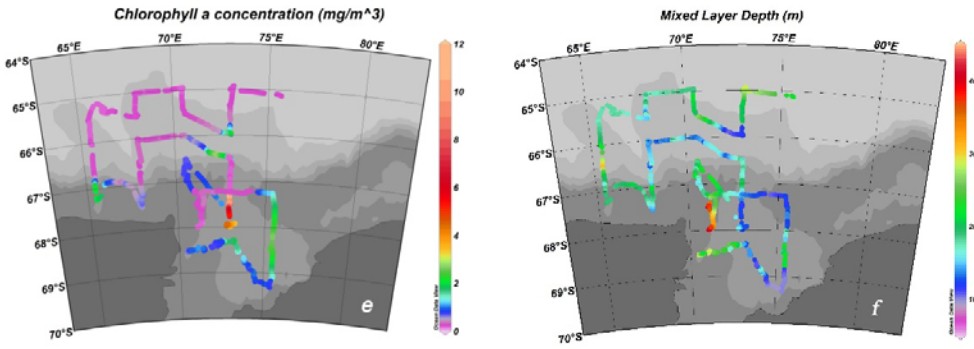

Fig.2 The distributions of underway oceanic and atmospheric $p$CO$_2$, SST, SSS, and CHL gridded from MODIS, as well as MLD gridded from station surveys, from February to early March.

## 2.2 SOM method and input variables

We hypothesize that oceanic $p$CO$_2$ can be reconstructed using the SOM method with four proxy parameters (Eq. 1): sea surface temperature (SST), chlorophyll-a concentration (CHL), mixed layer depth (MLD), and sea surface salinity (SSS).

$$p\text{CO}_2^{\text{sea}} = \text{SOM (SST, CHL, MLD, SSS)} \qquad (1)$$

The SOM is trained to project the input space of training samples to a feature space (Kohonen, 1984), which is usually represented by grid points in two-dimensional space. Each grid point, which is also called a neuron cell, is associated with a weight vector having the same number of components as the vector of the input data (Zeng et al., 2017). During SOM analysis, three steps are taken following Nakaoka et al. (2013) to estimate the oceanic $p$CO$_2$ fields (see Fig. 3). Because the four input environmental parameters (SST, CHL, MLD, and SSS) are used to estimate $p$CO$_2$ in this study, each input data set is prepared in 4-D vector form. Here, the SOM analysis was carried out using the MATLAB SOM tool box 2.0 (Vesanto, 2002). It has been developed by the

Laboratory of Computer and Information Science in the Helsinki University of Technology and is
available from the following web page: http://www.cis.hut.fi/projects/somtoolbox.

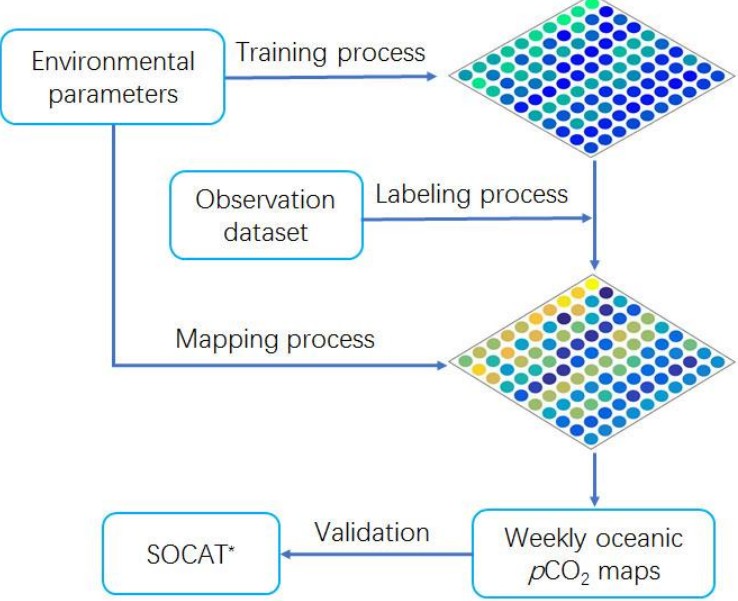


Fig. 3. Schematic diagram of the main three steps involved in the SOM neural network calculations used to
obtain weekly $pCO_2$ maps for February to early March of 2015.
During the training process, each neuron's weight vectors ($P_i$), which are linearly initialized,
are repeatedly trained by being presented with the input vectors ($Q_j$) of environmental parameters
in the SOM training function. Because SOM analysis is known to be a powerful technique with
which to estimate $pCO_2$ based on the non-linear relationships of the parameters (Telszewski et al.,
2009), we assumed that the non-linear relationships of the proxy parameters are sufficiently
represented after the training procedure. During this step, Euclidean distances (D) are calculated
between the weight vectors of neurons and the input vectors as shown in Eq.2, and the neuron with
the shortest distance is selected as the winner. This process results in the clustering of similar
neurons and the self-organization of the map. The observed oceanic $pCO_2$ data are not needed in
the first step.
$$D(\mathbf{P}_i, Q_j) = \sqrt{\left(P_{i\_SST} - Q_{j\_SST}\right)^2 + \left(P_{i\_CHL} - Q_{j\_CHL}\right)^2 + \left(P_{i\_MLD} - Q_{j\_MLD}\right)^2 + \left(P_{i\_SSS} - Q_{j\_SSS}\right)^2}$$

… (Eq. 2)

During the second part of the process, each preconditioned SOM neuron is labelled with an
observation dataset of in situ oceanic $pCO_2$values, and the labelling process technically follows
the same principles as the training process. The labelling dataset, which consists of the observed
$pCO_2$ and normalized SST, CHL, MLD and SSS data, is presented to the neural network. We
calculated the D values between trained neurons and observational environmental data sets. The
winner neuron is selected as in step1 and labelled with an observed $pCO_2$ value. After the labelling
process, the neurons are represented as 5-D vectors.
Finally, during the mapping process, the labelled SOM neurons created by the second process
and the trained SOM neurons created by the first process are used to produce the oceanic $pCO_2$
value of each winner neuron based on its geographical grid point in the study area.
Before the training process, the input training dataset and labelling dataset are analysed and
prospectively normalized to create an even distribution. The statistics and ranges of the values of
all variables are presented in Table 1. When the datasets of the four proxy parameters were
logarithmically normalized, the skewness values of CHL and MLD changed, especially for the
training dataset. The $N$ coverage represents the percentage of the training data that are labelled.
The data $N$ coverage values of the training data sets of CHL, MLD and SSS are 82.1%, 85% and
81.1%, respectively, which may be due to their insufficient spatiotemporal coverage and/or bias
between the labelling and training data sets. The $N$ coverage of the logarithmic datasets changed to
93.6% and to 98.7%for CHL and MLD, respectively. Thus, the common logarithms of the CHL
and MLD values are used for both the training and labelling datasets to resolve the data coverage
issue arising from significantly increasing the data coverage as well as to overcome the weighting
issue arising from the different magnitudes between variables (Ultsch and Röske, 2002).
Table 1. Statistics of labelling and training data sets showing the distribution and coverage of each
variable.

| Coverage of each variable | | SST[C] | CHL[mg/m$^3$] | MLD[m] | SSS[psu] |
|---|---|---|---|---|---|
| Labelling | Max | 0.81 | 11.13 | 40.69 | 33.81 |
| | Min | -1.44 | 0.17 | 7.84 | 32.43 |
| | Mean | -0.27 | 3.80 | 14.41 | 33.27 |
| | Skewness | 0.4(-0.2)[#] | 0.8(-0.3) | 0.9(0.4) | 0.6(0.6) |
| Training | Max | 2.48 | 40.17 | 48.95 | 34.17 |
| | Min | -1.8 | 0.06 | 10.46 | 28.64 |
| | Mean | -0.53 | 1.36 | 14.79 | 33.16 |
| | Skewness | 0.5(-0.6) | 4.3(0.5) | 2.6(0.8) | -0.9(-1.0) |
| | N coverage* (%) | 91.3(92.5)[+] | 82.1(93.6) | 85.0(98.7) | 81.1(80.4) |

# The skewness of the common logarithm of each variable is shown in parentheses.

 * [number of training data within the labelling data range]/[total number of training data]

 + The percent labelling data coverage of normalized variables is shown in parentheses

In this study, we construct weekly oceanic $pCO_2$ maps from February to early March of 2015 using four datasets, i.e., SST, CHL, MLD, and SSS. Considering the size of our study region, we chose a spatial resolution of 0.1° latitude by 0.1° longitude. For SST, we used daily data from AVHRR ONLY (https://www.ncdc.noaa.gov/oisst) with a 1/4° spatial resolution (see Fig.S1). CHL data represent the 8-D composite chlorophyll-a data from MODIS-Aqua (http://oceancolor.gsfc.nasa.gov) with a spatial resolution of 4 km (see Fig.S2). We also used the daily SSS and MLD data (see Fig.S3-4) from the 1/12° global analysis and forecast product from the Copernicus Marine Environment Monitoring Service (CMEMS, http://marine.copernicus.eu/). Sea ice concentration data are from the daily 3.125-km AMSR2 dataset (Spreen et al., 2008, available on https://seaice.uni-bremen.de, see Fig.S5).

All daily datasets were first averaged to 8-day fields, which are regarded as weekly in this study. The period from the beginning of February to early March comprises four independent week series: week-1 (from 02/02/2015 to 02/09/2015), week-2 (from 02/10/2015 to 02/17/2015), week-3 (from 02/18/2015 to 02/25/2015), and week-4 (from 02/26/2015 to 03/05/2015). The weekly proxy parameters (**SCMS**) were further re-gridded to a horizontal resolution of 0.1°·0.1° using the Kriging method in SURFER software (version 7.3.0.35). In the SOM analyses, input vectors with missing elements are excluded. We compared the assimilated datasets of SST from AVHRR with the in situ measurements obtained by CTD along the cruise. Their correlation is 0.97, and their root-mean-square error (RMSE) is 0.2°C. Comparing the SSS and MLD fields from the Global Forecast system with the in situ measurements yields correlations of 0.76 and 0.74 and RMSEs of 0.41 psu and 5.15 m, respectively. The uncertainty of the MODIS CHL data in the Southern Ocean is approximately 35% (Xu et al., 2016). For the labelling procedure, the observed oceanic $pCO_2$ together with the corresponding in situ SST, SSS, MLD, and MODIS CHL products in vector form are used as the input dataset.

**2.3 Validation of SOM-derived oceanic $pCO_2$**

More realistic $pCO_2$ estimates are expected from SOM analyses when the distribution and variation ranges of the labelling variables closely reflect those of the training data sets (Nakaoka et al., 2013). However, our underway measurements of $pCO_2$ values have spatiotemporal limitations preventing them from covering the range of variation of the training data sets. To validate the

oceanic $p$CO$_2$ values reconstructed by the SOM analysis, we used the fugacity of oceanic CO$_2$
datasets from the Surface Ocean CO$_2$ Atlas (hereafter referred to as "SOCAT" data,
http://www.socat.info) version 5 database (Bakker et al., 2016).We selected the dataset from
SOCAT(the EXPOCODE is 09AR20150128, see cruise in Fig. 4a) that coincided with the same
period as our study. The cruise lasted from Feb. 6 to Feb. 27, 2015, and $f$CO$_2$ measurements were
made every 1 min at a resolution of 0.01°. We recalculated $p$CO$_2$ values based on the obtained
$f$CO$_2$ values provided by the SOCAT data using the fugacity correction (Pfeil et al., 2013).

**2.4 Carbon uptake in the Prydz Bay**
The flux of CO$_2$ between the atmosphere and the ocean was determined using $\Delta p$CO$_2$ and the
transfer velocity across the sea-air interface, as shown in Eq. 3, where K is the gas transfer
velocity (in cm h$^{-1}$), and the quadratic relationship between wind speed (in units of m s$^{-1}$) and the
Schmidt number is expressed as (Sc/660)$^{-0.5}$. L is the solubility of CO$_2$ in seawater (in mol litre$^{-1}$
atm$^{-1}$) (Weiss, 1974). For the weekly estimation in this study, the scaling factor for the gas transfer
rate is changed to 0.251 for shorter time scales and intermediate wind speed ranges (Wanninkhof,
2014). Considering the unit conversion factor (Takahashi et al., 2009), the weekly sea-air carbon
flux in the Prydz Bay can be estimated using Eq. (4):

$Flux_{\text{sea-air}} = K \times L \times \Delta p\text{CO}_2$        (3)

$Flux_{\text{sea-air}}[\text{g C/( m}^2 \cdot \text{week)}] = 30.8 \times 10^{-4} \times U^2 \times (p\text{CO}_2^{\text{sea}} - p\text{CO}_2^{\text{air}})$     (4)

where U represents the wind speed 10 m above sea level, and $p$CO$_2^{\text{sea}}$ and $p$CO$_2^{\text{air}}$ are the partial
pressures of CO$_2$ in sea water and the atmosphere, respectively.
We downloaded weekly ASCAT wind speed data (http://www.remss.com/, see Fig. S6)
with a resolution of1/4° and then gridded the dataset to fit the 0.1° longitude · 0.1° latitude spatial
resolution of the SOM-derived oceanic $p$CO$_2$. We gridded the atmospheric $p$CO$_2$ data collected
along the cruise track to fit the spatial resolution of the SOM-derived oceanic $p$CO$_2$ data using a
linear method. The total carbon uptake was then obtained by accumulating the flux of each grid in
each area according to Jiang et al. (2008) and using the proportion of ice-free areas (Takahashi et
al., 2012). When the ice concentration is less than 10% in a grid, we regard the grid box as
comprising all water. When the ice concentration falls between 10% and 90%, the flux is
computed as being proportional to the water area. In the cases of leads or polynyas due to the
dynamic motion of sea ice (Worby et al., 2008), we assume the grid box to be 10% open water
when the satellite sea ice cover is greater than 90%.

**3 Results and discussion**

**3.1 The distributions of underway measurements**

During austral summer, daylight lasts longer and solar radiation increases. With increasing
sea surface temperature, ice shelves break and sea ice melts, resulting in the stratification of the
water column. Starting in the beginning of February, the R/V Xuelong sailed from east to west
along the sea ice edge, and its underway measurements are shown in Fig.2. Based on the water
depth and especially the different ranges of oceanic $p\mathrm{CO_2}$ (see Fig.2a and Table2), the study area
can be roughly divided into three regions, namely, the Open-ocean region, Sea-ice region and
Shelf region (see Table2).
The Open-ocean region ranges northward from 66°S to 64°S, where the Antarctic Divergence
Zone is located and water depths are greater than 3000 m. In the Open-ocean region, the oceanic
$p\mathrm{CO_2}$ was the highest, varying from 291.98 µatm to 379.31 µatm, with a regional mean value of
341.48 µatm. The Antarctic Divergence Zone was characterized by high nutrient concentrations
and low chlorophyll concentrations, with high $p\mathrm{CO_2}$ attributed to the upwelling of deep waters,
thus suggesting the importance of physical processes in this area (Burkill et al., 1995; Edwards et
al., 2004). The underway sea surface temperatures in this region are relatively high, with an
average value of -0.23°C due to the upwelling of Circumpolar Deep Water (CDW), while at the
sea ice edge (73°E, 65.5°S to 72°E, 65.8°S), the SST decreased to less than-1°C. From 67.5°E
westward, affected by the large gyre, cold water from high latitudes lowered the SST to less than
0°C. Near the sea ice edge, SSS decreased quickly to 31.7 psu due to the diluted water; along the
65°S cruise, it reached 33.3 psu; then, moving westward from 67.5°E, affected by the fresher and
colder water brought by the large gyre, it decreased to 32.5 psu. The satellite chlorophyll-a image
showed that the regional mean was as low as 0.45 mg/m$^3$, except when the vessel near the sea ice
edge recorded CHL values that increased to 2.26 mg/m$^3$. The lowest $p\mathrm{CO_2}$ value was found near
the sea ice edge due to biological uptake. The distribution of MLD varied along the cruise. Near
the sea ice edge, because of the melting of ice and direct solar warming, a low-density cap existed
over the water column, and the MLD was as shallow as 10.21 m. The maximum value of MLD in
the Open-ocean region was 31.67 m. In the Open-ocean region, atmospheric $p\mathrm{CO_2}$ varied from
374.6 µatm to 387.8 µatm. Along the 65°E cruise in the eastern part of the Open-ocean region, the
oceanic $p\text{CO}_2$ was relatively high, reaching equilibrium with atmospheric $p\text{CO}_2$. In the western
part of this region, the oceanic $p\text{CO}_2$ decreased slightly due to the mixture of low $p\text{CO}_2$ from
higher latitudes brought by the large gyre. Mixing and upwelling were the dominant factors
affecting the oceanic $p\text{CO}_2$ in this region.
The seasonal Sea-ice region (from 66°S to 67.25°S) is located between the Open-ocean region
and the Shelf region. In this sector, sea ice changed strongly, and the water depth varied sharply
from 700 m to 2000 m. The oceanic $p\text{CO}_2$ values ranged from 190.46 µatm to 364.43 µatm, with a
regional mean value of 276.48 µatm. Sea ice continued to change and reform from late February to
the beginning of March (Fig. 6). The regional mean sea surface temperature decreased slightly
compared to that in the Open-ocean region, and the average value was -0.72°C. With the rapid
changes in sea ice, the sea surface temperature and salinity varied sharply from -1.3°C to 0.5°C
and from 31.8 psu to 33.3 psu, respectively. When sea ice melted, the water temperature increased,
biological activity increased, and the chlorophyll-a value increased slightly to reach a regional
average of 0.59 mg/m$^3$. Due to the rapid change in sea ice cover, the value of MLD varied from
12.8 m to 30.9 m.
The Shelf region (from 67.25°S southward) is characterized by shallow depths of less than
700 m, and it is surrounded by the Amery Ice Shelf and the West Ice Shelf. Water inside the Shelf
region is formed by the modification of low-temperature and high-salinity shelf water (Smith et al.,
1984). The Prydz Bay coastal current flows from east to west in the semi-closed bay. The oceanic
$p\text{CO}_2$ values in this region were the lowest of those in all three sectors; these values ranged from
151.70 µatm to 277.78 µatm, with a regional average of 198.72 µatm. A fresher, warmer surface
layer is always present over the bay, which is known as the Antarctic Surface Water (ASW).
During our study period, the Shelf region was the least ice-covered region. A large volume of
freshwater was released into the bay, resulting in low sea surface temperature (an average of
-0.61°C) and salinity (an average of 32.4 psu) values. As shown in Fig.2f, the mixed layer depth in
most of the inner shelf is low. Due to the vast shrinking of sea ice and strong stratification in the
upper water, algal blooming occurred and chlorophyll values were high, with an average of 1.93
mg/m$^3$. The chlorophyll-a value was remarkably high, reaching11.04 mg/m$^3$when sea ice retreated
eastwardly from 72.3°E, 67.3°S to 72.7°E, 68°S. The biological pump became the dominant
factor controlling the distribution of oceanic $p\text{CO}_2$. In the bay mouth close to the Fram Bank, due
to local upwelling, the water salinity increased remarkably to approximately 33.2 psu.
Table2 The regional mean values of underway measurements in three sub-regions

| | $p$CO$_2$ [μatm] | SST [°C] | CHL [mg/m$^3$] | MLD [m] | SSS [psu] |
|---|---|---|---|---|---|
| Open-ocean region (66°S - 64°S) | 341.48 | -0.23 | 0.45 | 20.13 | 32.61 |
| Sea-ice region (66°S - 67.25°S) | 276.48 | -0.72 | 0.59 | 19.44 | 32.42 |
| Shelf region (67.25°S - 70°S) | 198.72 | -0.61 | 1.95 | 16.84 | 32.46 |


## 3.2 Quality and maps of SOM-derived oceanic $p$CO$_2$

We selected SOM-derived oceanic $p$CO$_2$ values to fit the cruise track of SOCAT for the same
period in February 2015 using a nearest-grid method. The RMSE between the SOCAT data and
the SOM-derived result was calculated as follows:
$$RMSE = \sqrt{\frac{\sum\left(pCO_2^{sea}(SOM) - pCO_2^{sea}(SOCAT)\right)^2}{n}} \qquad (4)$$
where n is the number of validation datasets. The RMSE can be interpreted as an estimation of the
uncertainty in the SOM-derived oceanic $p$CO$_2$ in the Prydz Bay. In this study, the RMSE of the
SOM-derived oceanic $p$CO$_2$ and SOCAT datasets is 22.14 μatm, and the correlation coefficient
$R^2$ is 0.82. The absolute mean difference is 23.58 μatm. The RMSE obtained in our study is
consistent with the accuracies (6.9 μatm to 24.9 μatm) obtained in previous studies that used
neuron methods to reconstruct oceanic $p$CO$_2$ (Nakaoka et al., 2013; Zeng et al., 2002; Sarma et al.,
2006; Jo Y H et al., 2012; Hales et al., 2012; Telszewshi et al., 2009). The precision of this study is
on the high side of those that have been previously reported. The slope of the scatter plot
indicates that the SOM-derived oceanic $p$CO$_2$ data are lower than the SOCAT data (see Fig. 4b).
Thus, the precision of these data may have greater uncertainty because the SOCAT dataset does
not cover the low-$p$CO$_2$ area towards the south. Thus, increasing the spatial coverage of the
labelling data will help increase the precision of the SOM-derived oceanic $p$CO$_2$.

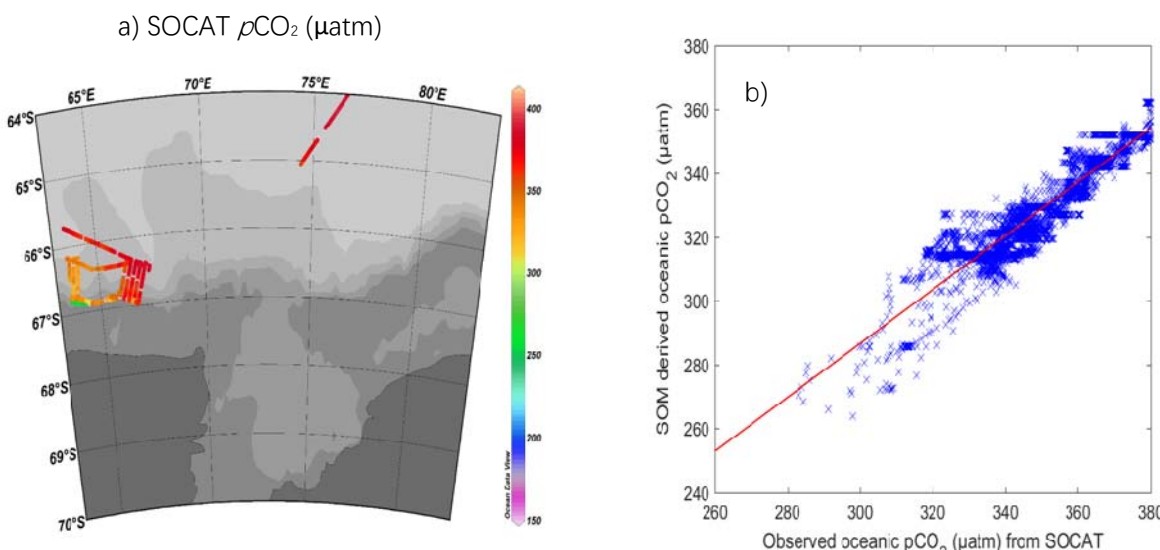

a) SOCAT $p$CO$_2$ (µatm)

Fig. 4 a) The cruise lines from SOCAT used to validate the SOM-derived oceanic $p$CO$_2$ for the study period in 2015; b) comparison between the SOM-derived and observed SOCAT oceanic $p$CO$_2$ data.

### 3.3 Spatial and temporal distributions of SOM-derived oceanic $p$CO$_2$

The weekly mean maps of SOM-derived oceanic $p$CO$_2$ in the Prydz Bay are shown in Fig. 5. In the Open-ocean region, the oceanic $p$CO$_2$ values were higher than those in the other two regions due to the upwelling of the CDW. During all four weeks, this region was nearly ice-free, while the average sea ice coverage was 18.14% due to the presence of permanent sea ice (see Fig.6). The oceanic $p$CO$_2$ distribution decreased from east to west in the Open-ocean region, with lower values observed at the edge of sea ice. In the western part of the Open-ocean region, oceanic $p$CO$_2$ decreased due to mixing with low oceanic $p$CO$_2$ flowing from high-latitude regions caused by the large gyre. From week-1 to week-4, the maximum oceanic $p$CO$_2$ increased slightly and reached 381.42 µatm, which was equivalent to the $p$CO$_2$ value of the atmosphere.

In the Sea-ice region, sea ice continued to rapidly melt and reform. The weekly mean sea ice coverage percentage was 29.54%, occupying nearly one-third of the Sea-ice region. As shown in Fig.5, the gradient of the oceanic $p$CO$_2$ distribution increased from south to north affected by the flow coming from the Shelf region by the large gyre. In the eastern part of this region, adjacent to the sea ice edge, the oceanic $p$CO$_2$ values were lower. The oceanic $p$CO$_2$ changed sharply from 155.86 µatm (near the sea ice edge) to 365.11 µatm (close to the Open-ocean region).

In austral winter, the entire Prydz Bay basin is fully covered by sea ice, except in a few areas,
i.e., the polynyas, which remain open due to katabatic winds (Liu et al., 2017). When the austral
summer starts, due to coincident high wind speeds, monthly peak tides, and/or the effect of
penetrating ocean swells, the sea ice in the Shelf region starts to melt first in early summer (Lei et
al., 2010), forming the Prydz Bay Polynya. The semi-closed polynya functions as a barrier for
water exchange in the Shelf region and causes a lack of significant bottom water production,
hindering the outflow of continental shelf water and the inflow of Antarctic circle deep water,
resulting in the longer residence time of vast melting water and enhanced stratification (Sun et al.,
2013). Due to vast melting of the sea ice, the sea surface salinity decreased and algae bloomed;
biological productivity promptly increased, and the chlorophyll-a concentration reached its peak
value. As shown in Fig. 5, the distribution of oceanic $p$CO$_2$ in the Shelf region was characterized
by its lowest values. The obvious drawdown of oceanic $p$CO$_2$ occurred in the Shelf region due to
phytoplankton photosynthesis during this summer bloom. The lowest oceanic $p$CO$_2$ in the Shelf
region was 153.83 μatm, except at the edge of the West Ice Shelf, where the Shelf oceanic $p$CO$_2$
exceeded 300 μatm. The oceanic $p$CO$_2$ was the lowest in week-1, which coincided with a peak in
chlorophyll-a, as evidenced by satellite images. The regional oceanic $p$CO$_2$ increased slightly in
week-4 compared to the other three weeks.

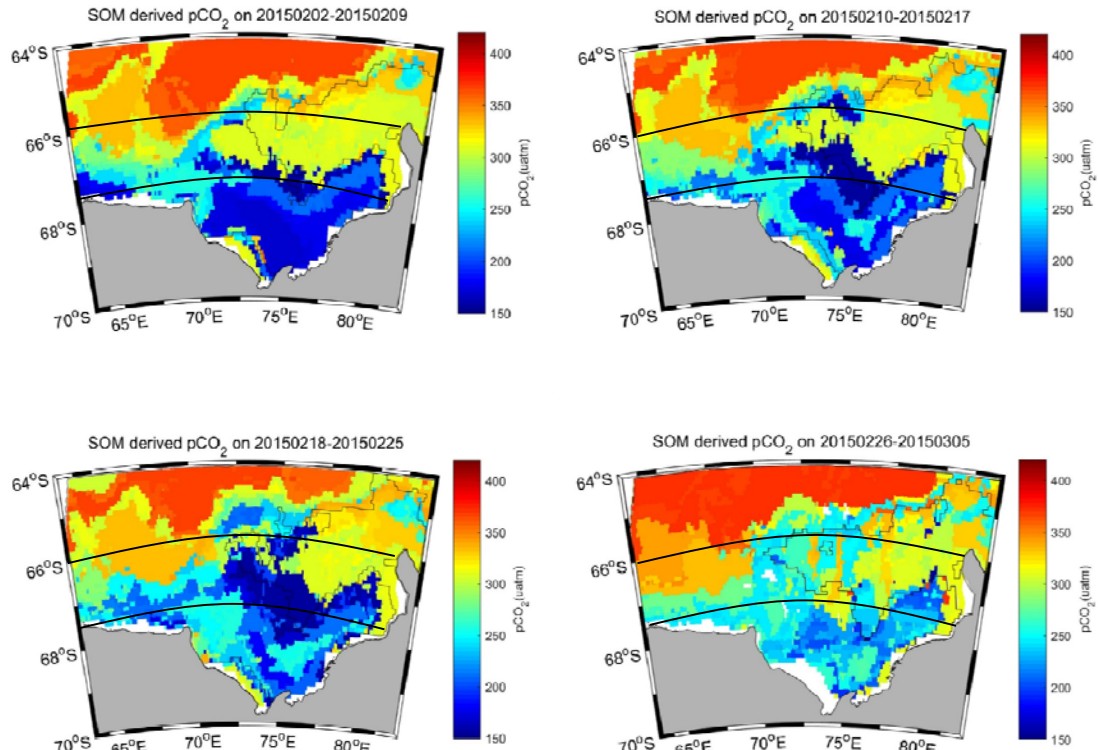


Fig.5 Distribution of weekly mean SOM-derived oceanic $p$CO$_2$ in the Prydz Bay (unit: µatm) from Feb. 2,
2015 to Mar. 5, 2015. The black contour represents a sea ice concentration of 15%.

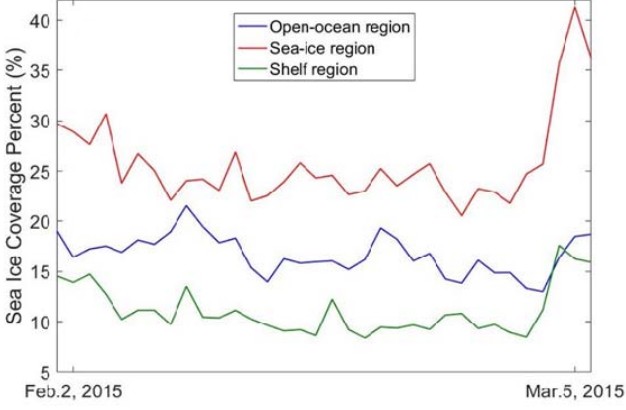


Fig. 6 Percentage of sea ice coverage in three sub-regions from Feb. 2, 2015 to Mar. 5, 2015 (blue:
Open-ocean region; red: Sea-ice region; green: Shelf region).
**3.4 Carbon uptake in the Prydz Bay**

During our study period, the entire region was undersaturated, with $CO_2$ being absorbed

by the ocean. The regional averaged ocean-air $pCO_2$ difference ($\Delta pCO_2$) was highest in the Shelf
region, followed by the Sea-ice region and Open-ocean region (see Table3). The regional and
weekly mean $\Delta pCO_2$ in the Shelf region changed from -184.31 µatm in week-1 to -141.00 µatm in
week-2 as chlorophyll decreased. The $\Delta pCO_2$ in the Sea-ice region and Open-ocean region
showed the same patterns, increasing from week-1 to week-3 and then decreasing in week-4.
Based on the $\Delta pCO_2$ and wind speed data, the uptake of $CO_2$ in these three regions is presented in
Table3.The uncertainty of the carbon uptake depends on the errors associated with the wind speed,
the scaling factor and the accuracy of the SOM-derived $pCO_2$ according to Eq.4. The scaling factor
will yield a 20% uncertainty in the regional flux estimation. The errors in the wind speeds of the
ASCAT dataset are assumed to be 6% (Xu et al., 2016); the error in the quadratic wind speed is
12%.The RMSE of the SOM-derived $pCO_2$ is 22.14 µatm. Considering the errors described above
and the uncertainty occurring when the sea-air computation expression is simplified (1.39%, Xu et
al., 2016), the total uncertainty of the final uptake is 27%. In the Shelf region, the low oceanic
$pCO_2$ levels drove relatively intensive $CO_2$ uptake from the atmosphere. The carbon uptake in the
Shelf region changed from week-1 (2.51±0.68 TgC, $10^{12}$ gram=Tg) to week-2 (2.77±0.75 TgC).
In contrast, in week-3, the wind speed slowed down, resulting in the uptake of $CO_2$ in the Shelf
region decreasing to 2.10±0.57 TgC. In week-4, even though the $\Delta pCO_2$ was the lowest of all four
weeks, the total absorption still increased to 2.63±0.715 TgC due to the high wind speed (with an
average value of 7.92 m/s). The total carbon uptake in the three regions of the Prydz Bay,
integrated from February to early March of2015, was 23.57 TgC, with an uncertainty of ±6.36
TgC.

Table3 Regional and weekly mean $\Delta pCO_2$, wind speed and uptake of $CO_2$ in three

sub-regions (negative values represent directions moving from air to sea).

| | | Week-1 | Week-2 | Week-3 | Week-4 | Uptake in 4 weeks[Tg] |
|---|---|---|---|---|---|---|
| Open-ocean region (66°S - 64°S) | $\Delta pCO_2$ [µatm] | -34.11 | -42.69 | -51.94 | -34.08 | |
| | Wind speed [m/s] | 7.82 | 8.54 | 7.02 | 9.31 | -5.74 |
| | Uptake [Tg] | -1.08 | -1.55 | -1.51 | -1.60 | |
| Sea-ice region (66°S - 67.25°S) | $\Delta pCO_2$[µatm] | -115.92 | -119.83 | -127.74 | -86.72 | |
| | Wind speed[m/s] | 7.67 | 8.17 | 6.39 | 8.36 | -7.82 |
| | Uptake [Tg] | -2.11 | -2.35 | -1.73 | -1.63 | |
| Shelf region (67.25°S - 70°S) | $\Delta pCO_2$[µatm] | -184.32 | -170.23 | -158.61 | -141.03 | |
| | Wind speed[m/s] | 6.92 | 7.27 | 6.67 | 7.92 | -10.01 |
| | Uptake [Tg] | -2.51 | -2.77 | -2.10 | -2.63 | |


Roden et al. (2013) estimated the coastal Prydz Bay to be an annual net sink for $CO_2$ of
$0.54\pm0.11$ mol/(m$^2$·year), i.e., $1.48\pm0.3$ g/(m$^2$·week). Gibsonab et al. (1999) estimated the average
sea-air flux in the summer ice-free period to be more than 30 mmol/(m$^2$·day), i.e., 9.2 g/(m$^2$·week).
Our study suggests that the sea-air flux during the strongest period of the year, i.e., February, could
be much larger. The average flux obtained here, 18.84 g/(m$^2$·week), is twice as large as the average
value estimated over a longer period (November to February) reported by Gibsonab et al. (1999).
As the region recording the strongest surface unsaturation of these three regions in summer,
the Shelf region has a potential carbon uptake of $10.01\pm2.7$ Tg C from February to early March,
which accounts for approximately 5.0‰-6.7‰ of the net global ocean $CO_2$ uptake according to
Takahashi et al. (2009), even though its total area is only $78*10^3$ km$^2$. Due to the sill constraint,
there is limited exchange between water masses in and outside the Prydz Bay. During winter, the
dense water formed by the ejection of brine in the Bay can potentially uptake more anthropogenic
$CO_2$ from the atmosphere that can descend to greater depths, thus enhancing the acidification in
deep water. According to Shadwick et al. (2013), the winter values of $p$H and $\Omega$ decrease more
remarkably than those in summer. As the bottom water in the Prydz Bay is a possible source of
Antarctic Bottom Water (Yabuki et al., 2006), the Shelf region may transfer anthropogenic $CO_2$ at
the surface to deep water and may thus influence the acidification of the deep ocean over long
timescales.

**4 Summary**
Based on the different observed ranges of the distribution of ocean $p$CO$_2$, the Prydz Bay
region was divided into three sectors from February to early March of 2015. In the Shelf region,
biological factors exerted the main control on oceanic $p$CO$_2$, while in the Open-ocean region,
mixing and upwelling were the main controls. In the Sea-ice region, due to rapid changes in sea ice,
oceanic $p$CO$_2$ was controlled by both biological and physical processes. SOM is an important tool
for the quantitative assessment of oceanic $p$CO$_2$ and its subsequent sea-air carbon flux, especially
in dynamic, high-latitude, and seasonally ice-covered regions. The estimated results revealed that
the SOM technique can be used to reconstruct the variations in oceanic $p$CO$_2$ associated with
biogeochemical processes expressed by the variability in four proxy parameters: SST, CHL, MLD
and SSS. The RMSE of the SOM-derived oceanic $p$CO$_2$ is 22.14 μatm for the SOCAT dataset.

From February to early March of 2015, the Prydz Bay region was a strong carbon sink, with a carbon uptake of 23.57±6.36 TgC. The strong potential uptake of anthropogenic $CO_2$ in the Shelf region will enhance the acidification in the deep-water region of the Prydz Bay and may thus influence the acidification of the deep ocean in the long run because it contributes to the formation of Antarctic Bottom Water.

**Acknowledgments**

This work is supported by National Natural Science Foundation of China (NSFC41506209,41630969, 41476172, 41230529), Qingdao National Laboratory for marine science and technology (QNLM2016ORP0109), Chinese Projects for Investigations and Assessments of the Arctic and Antarctic (CHINARE2012-2020 for 01-04, 02-01, and 03-04). This work is also supported by Korea Polar Research Institute grants PE19060 and PE19070. We would like to thank China Scholarship Council (201704180019) and State Administration of Foreign Experts Affairs P. R. China for their support in this research. We would like to thank the carbon group led by Zhongyong Gao and Heng Sun in GCMAC and the crew on R/V Xuelong for their support on the cruise. We are thankful to contributors of the SOCAT database for validated $p$CO2 data and Mercator Ocean for providing the Global Forecast model output. We deeply appreciate Dr. Xianmin Hu in Bedford Institute of Oceanography, who provided us with useful technical instructions.

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
