# Peer review of "Variations in the Summer Oceanic pCO2 and Carbon Sink in the Prydz Bay Using the SOM Analysis Approach 2 Suqing Xu1, Keyhong Park2\*, Yanmin Wang3, Liqi Chen1\*, Di Qi1, Bingrui Li4 3 Key Laboratory of Global Change and"

_Biogeosciences, 2018_

## Referee Comment (RC1) · Anonymous Referee #1 · 6 Aug 2018

Summary:

Xu and colleagues investigate the regional sea surface pCO2 and air-sea flux in Prydz Bay Antarctica using observations from the CHINARE cruise in February 2015. The authors divide the study regions into 3 sub-regions, based on the physical and bio-geochemical controls of these sub-regions. Using a self-organizing map approach, the authors extrapolate the cruise data to the entire study region in order to estimate the carbon exchange of Prydz Bay.

The Southern Ocean is still among the least observed and certainly least well understood ocean basins, hence I found this process study – investigating carbon variability

and air-sea exchange in Prydz Bay - to be very interesting and certainly relevant for the BG readership. More details on the strengths and weaknesses are listed below.

Strengths:

I found the manuscript and particularly the discussion of the processes comprehensive and logically built-up. The authors further make use of an appropriate and previously applied method based on machine learning (i.e. the SOM method) to extrapolate the cruise information to the full region of interest. They use independent validation data to test how well their approach reproduces observations from the SOCAT dataset and use this information to estimate the uncertainty of their integrated air-sea flux.

Weaknesses:

Up-front, I would like to note that there are several language issues – too many to be all named here (just one examples: line 232: "In Pacific Ocean" should be "In the Pacific Ocean") – hence I do recommend English language editing.

During my review, I have encountered a few things that need clarification or some more information from the authors. They are listed from the most to least concerning. Additional comments (not of major concern) with line-numbers can be found at the end of this document:

.)Method section: At the moment, it is impossible for a reader who has not worked with the SOM approach to understand the methods section. Sentences like:" The SOM is trained using unsupervised learning to project the input space of training samples to a feature space (Kohonen, 1984), which is usually represented by grid points in two-dimension space." Imagine a BG reader who is interested in the carbon exchange of Prydz Bay but has never worked with a SOM. How is that person supposed to understand wording like "unsupervised, feature space, weight vector, training data, labelling data, etc." without reading several other papers first? As a SOM user I had no issues to follow this section but in my view, it has to be simplified for the more general BG

audience. Furthermore, the authors miss to mention what distance function the SOM uses to detect the "winner neuron" (Euclidean distance maybe?). Furthermore, I don't think the phrase "resolve nonlinear relationships" (see abstract) is appropriate, since a SOM is a clustering algorithm that clusters based on similarities, but does not explicitly "resolve" a relationship.

.)Training data: This links a bit to my point above but goes a bit more in-depth: I am not sure how data have been handled. On line 177 the authors state that the data have been "the four proxy parameters were logarithmically normalized" but table 1 suggests otherwise. In table 1 all values are absolute values. Besides that, I am not convinced that it makes sense to logarithmically normalize all 4 proxies. It makes sense for the skewed MLD and CHL-a but not really for salinity and temperature. Besides, I wonder how the normalization effects the distance function (which is not mentioned). Euklidian distances depend on the data-value range of each proxy. Also, what I am missing is a discussion why exactly the 4 proxies have been chosen? Why not sea surface height, wind speed, sea level pressure? What makes the 4 proxies so unique? I know they have been used by other authors, but the reader of THIS study needs this information.

.)Uncertainty: line 389 states: "increased from week-1 (2.13 TgC) to week-2 (2.24 TgC) due to increased wind speed." I was a bit disappointed here. First there is the effort to calculate uncertainties, then it is neglected in the text. Given the final uncertainty estimate, it is very unlikely that this regional difference of 0.1 TgC is significant. In general I suggest to add uncertainties wherever possible to avoid such misinterpretations.

.)Validation, comparison: I appreciate that the authors do a comparison with SOCAT data and include this in the overall flux uncertainty. I think that there need to be a bit more info in the text what cruise from SOCAT you are comparing to (this information is available on socat.info), or what the average spatial and temporal distance (which should be possible since a nearest grid method was used) between the cruises is. That certainly contributes to the mismatch as well. Otherwise, I was quite impressend by the relatively small ($\sim22\mu$atm) difference. It might not sound small at first but you are

comparing small special scale and high frequency temporal scale data based on the extrapolation of a single cruise. Therefore, 22$\mu$atm is impressive in my view. Furthermore, the RMSE tells the reader about the spread, but it would be valuable to add the mean (or absolute mean) difference between the SOM derived CO2 and the SOCAT cruise. This would give you an indication of the bias.

.)Methods section: On many occasions the authors re-grid data to the desired 0.1x0.1° resolution, but a bit more information on all data that were regridded and the algorithm would be appreciated. Ideally in form of a table. Additionally, I am missing the motivation why 0.1x0.1° was chosen. Why not 0.5x0.5 or even 0.05x0.05. Just to be clear, I don't suggest changing the resolution, but the text needs some motivation/technical explanation on why the current resolution was chosen that justifies all the data handling (i.e. regridding of proxy data)

Recommendation:

I have found this study to be interesting and to be of value to the BG readership. While I have raised some (partly major) concerns above I think that they can be resolved by the authors. I therefor recommend major revisions of the manuscript.

Specific and minor comments to the text:

Abstract line 14: Please also add the temporal resolution to the spatial resolution

Abstract lines 27-29: This last sentence is out of context and is not something you can conclude from this study, hence it needs to be removed.

Lines 32-33 reads "The role of the ocean south of 60S in the transport of CO2 to or from the atmosphere is still uncertain despite of its importance of reducing anthropogenic CO2 in the atmosphere" – that is a conflicting statement as it currently reads. If we know the importance of reducing atmospheric CO2 how can its role be uncertain?

Lines 76-77: "Therefore, the direction of the sea-air CO2 transfer is mainly regulated by the oceanic pCO2" – this statement needs a reference

Line 84: "The SOM analysis, based on neural network (NN), a type of artificial neural network" – the second part (based on neural network) can be removed

Line 117: "Salinity records the physical processes" – When I read this sentence I also think of larger scale circulation and mixing in the context of physical processes, whereas this statement links to the follow-up discussion about brine rejection. Maybe a different term would be more appropriate.

Line 130: How was the interpolation done?

Lines 133-136: "The mixed layer links the atmosphere to the deep ocean and plays a critical role in climate variability. Very few studies have emphasized the importance of accounting for the vertical mixing through the mixed layer depth" – Firstly, I disagree. Several studies have emphasized the importance of vertical mixing of carbon (but also nutrients, etc) through the mixed layer. Secondly, I caution the authors to mention the role in climate variability here. Their study does not resolve the necessary timescales to discuss either seasonal or interannual or decadal (whatever variability the authors refer to) variability.

Lines 154-155 "SOM based multiple non-linear regression" – This must have been a mistake or typo here, since the SOM (unlike e.g. a backpropagation network) does not perform a regression (also not a non-linear one). Instead the SOM clusters data based on similar environmental conditions.

Lines 194-195: "until the neural network sufficiently represents the nonlinear interdependence of proxy parameters used in training." – how is this judged? When do you know that its sufficient? I suppose this is judged by the number of SOM iterations, but how is set?

Line 215: "I could not figure out where the factor 30.8X10-4 comes from? Please explain in the text

Line 264: "robustly divided" – I caution the authors here: How can you be sure the

division is "robust"? Have you done any test that would proof robustness?

Lines 281-282: "region atmospheric pCO2 was stable from 374.6 $\mu$atm to 387.8 $\mu$atm" That is a difference of 13$\mu$atm – I would not call this stable at all! I suppose this difference is largely the result of sea level pressure variability and relative humidity in the surface layer, hence it would be interesting to see the molar fractions (in ppm) for comparison if available.

Line 285: "biological consume" – should be "biological uptake"

Line 318-319: "for a same period" – This would be important information. Furthermore, have you considered ARGO biogeochemistry floats from the SOCCOM array? They are deployed since 2013 and may add some additional independent estimate. This might however be beyond this manuscript.

Figure 4b: "It would be easier visible if x-axis an y-axis range would be the same.

---

## Referee Comment (RC2) · Anonymous Referee #2 · 7 Sep 2018

**General comments**

The manuscript 'variation of Summer Oceanic $pCO_2$ and Carbon Sink in the Prydz Bay Using SOM Analysis Approach' by Suqing Xu et al. presents their cruise data plus its analysis regarding oceanic and atmospheric $pCO_2$ and the related air-sea $pCO_2$ flux. The results can potentially be of interest to readers interested in the Southern Ocean carbon cycling, and its variability in time and space. It also provides an opportunity to the authors to show a practical example of the application of SOM in biogeochemistry. In order for the manuscript to be appreciated by the biogeochemical community, the authors should provide a better description of its relevance and importance for the greater Southern Ocean. As I am not an expert on SOM or neural networks, I cannot judge the methodology on that method in detail. I should however be able to understand what is presented in section 2.2, and I find this difficult at times. Several times mention is made of methods (like 'a linear method' or 'Linear regression extrapolation method') without further information on what is done: This makes reproducibility of the work without consulting the authors impossible. Besides that, I unfortunately often find the language to be confusing/imprecise, and therefore recommend professional English language checking before resubmitting. The language made it more difficult for me to judge the value of the manuscript, and I expect I can provide a more in-depth review after the language is improved. The manuscript would also improve if it were shortened as compared to the current version, as there is enough space to increase the information density in the manuscript in my opinion.

**Specific comments**

1. The introduction
   The introduction thoroughly describes the geographic setting of the Prydz Bay. I appreciate this, but it makes the introduction unbalanced as the questions 'why is this study of relevance' and 'what is new' are only covered by a few sentences. The authors describe the issue that the manuscript wants to address, namely the sparse spatiotemporal coverage of the Southern Ocean (SO) carbon cycle. They also tell the reader that they address the issue using the SOM approach. However, to what extent does research on the Prydz Bay support our understanding of the SO carbon cycle? On page 2, line 38-39 it is mentioned that the Prydz Bay is the third largest embayment in the Antarctic continent. No other reasons are given for the study of in specific this bay: What makes this bay (potentially) important for the SO carbon cycle even though it is small as compared to the total surface area of the SO? To what extent is this Bay representative for the SO as a whole (or just other parts of the SO), i.e. do the authors think their approach or data are useful for and representative of other areas in the SO? Why was the month February chosen to do the cruise?
   In the first sentence, it is mentioned that the SO is important for anthropogenic $CO_2$ uptake. The authors cannot distinguish between natural and anthropogenic carbon fluxes based on their measurements: Some sentences should be added to describe that the SO is a natural source of carbon to the atmosphere, but a sink for anthropogenic carbon – and that both are highly variable but creating a net sink for total carbon over the past decades. Here an argument could be made for their own study and cruise, which aims to reduce the spatiotemporal sparsity of the data and get a better understanding of the variability of the contemporary $\Delta pCO_2$ and its driving mechanisms. The authors call the Bay a sink at several instances (for example P4, l101 and P5, l125): Some numbers from previous studies should

be given to support the statement that the Bay as a whole is a sink for carbon before presenting your own results.

In Figure 1, an inset could be added to visualise the location of Fig. 1 on the Antarctic continent.

P3, l64-66: How does a marine ecosystem interact with the physical environment to make it complicated to study pCO2? Clarify your statement, as it currently is imprecise.

When describing the methods, clarify that in situ data from the cruise are combined with remotely sensed data to arrive at a gridded product.

2.1 In situ data

Here the authors present how they took their underway measurements and present them in Fig. 2. The first time I read this section, I missed a good structure: The section starts with an explanation of the cruise and instruments used (until line 115). Then, the following paragraphs came to me as a bit of a surprise. One could help the reader find a better flow through the text by explaining that there are several processes/water characteristics that can influence the pCO2 flux (which is the topic of this study). Then, the sea ice paragraph (lines 116-120), the information on the SSS and SST collection (lines 120-124), the biology/CHL paragraph (lines 125-131), and the MLD paragraph (lines 132-end of section) come more naturally. It is important to defend why specifically these proxies/data are used to do your study (create a gridded pCO2 map). Don't forget to start the title with a capital letter i. It is unclear to me whether the results presented in Fig. 2 are 4-week mean results or how they are calculated from the 4 cruise legs: Add more information to both the caption and the text.

2.2 SOM method and input variables

This section is generally hard to follow, maybe partly because I am not familiar with SOM. It should be improved so that also people new to SOM are able to understand and appreciate what you have done. Which 'environmental parameters' and which 'observational datasets' (Fig. 3) are used? Lines 205-220(or even up to 228) could be moved up in order to introduce the reader earlier to the datasets. Then the authors can explain what they are used for and how.

2.3 Validation of SOM derived oceanic $p$CO$_2$

This section raises a lot of questions from my side. To what extent is SOCAT comparable to your data? Are the data both summer data? Why do you talk about assimilating several years together, but then only take 2015 from SOCAT (line 239)? Could you maybe compare your data to a model estimate of $p$CO$_2$ for this region? Lines 232-235: How is the equilibrium used to assimilate a dataset over different years? There is generally no equilibrium between atmospheric and surface ocean $p$CO$_2$, do you mean $p$CO$_2$-disequilibrium? Why do you describe this if you did not apply this method after all?

2.4 Carbon uptake in the Prydz Bay

This section is quite clear to me: You have combined wind speed data and your $p$CO$_2$ measurements to arrive at a flux using Eq 2. However, you should clarify 1) where you used a 'scaling factor' (P 10, l247-248) (in Eq. 2?), and 2) that that you used your SOM-based $p$CO$_2$ product to calculate $\Delta p$CO$_2$ in Eq. 2 (did you?). In addition, you write that the transfer velocity is a function of wind speed and temperature (line 245) and then you write about a

gas transfer rate (line 248) (=transfer velocity?) which you apply a scaling factor to. I am left with the question which gas transfer rate or velocity you have used / how you calculated it.

3.1 the distribution of underway measurements
Here you present your underway measurements for three areas. On what basis did you divide the Prydz bay in these subregions? You write the division is 'robust' (P11, l264): Did you test what effect the choice of your division has on your results? It would be helpful to the reader if you added a plot figure with the subdivision of the Prydz bay into its three regions. Add units to all numbers (especially salinity lacks the psu unit throughout this section). I assume you are describing the results that are visualized in Fig 2 in this section: you should make reference to it if this is the case. Throughout the text of this section, you should be more precise on whether the values are regional means, 4-week means, and how you calculated this (refer to the methods). When you say decrease or increase (like P12, l291), it is not always clear to me whether it decreases/increases in time or space or whether the mean is lower or higher than in the neighbouring sub-region. This causes for example confusion when SST's 'vary sharply' (l 293) but 'decreased slightly' just the sentence above (l 291). The readability of this section may improve by summarizing your main results in a table. A sentence should be added either here on the methods where the relationship between chlorophyll-a (as remotely observed) and biological productivity is stated.

3.2 Quality and maps of SOM-derived oceanic $pCO_2$
You compare your results to SOCAT and calculate the RMSE. Could you also provide the $R^2$ of the best-fit line (red line in Fig. 4b)? You say your RMSE is consistent but not as good as most of the neuron methods. Do you mean it is on the high side of the accuracies previously reported, or why is it not as good? Could you calculate/estimate how many extra data points you would need to gain an improved precision of your SOM approach? You could probably comment on the limited amount of data that retrieving more data is not realistic with the resources and time available. SOCAT is not perfect either: A comment on its limited overlap with your study area would be appropriate here. It is surprising that the SOM estimate is generally higher than the SOCAT one, as SOCAT does not cover the low-$pCO_2$ area towards the south. Did you sample your SOM-derived $pCO_2$ dataset on the SOCAT locations, or did you compare all SOCAT in the area to all your data points in Fig. 4b? The first would probably be a fairer comparison and provide a better outcome as well. Fig. 4a could be plotted in the same way as Fig. 2 to make it easier for the reader to compare the spatial coverage.

3.3 Spatial and temporal distributions of SOM-derived $pCO_2$
Here I expect the presentation of your main result: the $pCO_2$ maps of Figure 6. However, the text mostly describes the sea ice situation of the region: Why is this done here? Maybe a different title would be more appropriate? If sea ice is a main driving factor for $pCO_2$, this should be argued using the results. If the authors could add regional sub-division lines on the maps in Fig. 6, it might be easier to argue for the chosen sub-division (i.e. Shelf region, etc.).

3.4 Carbon uptake in Prydz Bay
This section is quite clear, although it would be good to clarify when mean values are reported, and whether they are regional means or temporal means, or both. From the figure on page 17 (which has no number?) it is hard to read the $\Delta pCO_2$ changes: One could either

present it as a table, or adjust the y-axis range. Please make sure the figure is suitable for the colour blind (and check this throughout the manuscript): Use for example different shapes for the three different lines in the upper graph, and add shapes in the lower one.

Supplementary information
The text at the start of the SI is already used in the main text, I do not see the need to provide it twice, and would recommend to remove it from the SI.

**Technical corrections**

I made an effort to pick out the most important language issues. However, as recommended in the general comments, I would strongly advise the authors to revise their language throughout the manuscript and to have it checked before resubmitting.

1. Try to prevent the use of the word 'it' throughout the manuscript: replace by the actual subject of the sentence
2. Caption of Fig. 1: replace 'The circulations in the' by 'The ocean circulation in the'. Replace sentence 'The weekly sea ice extents for our study periods were overlapped on the cruise.' by 'During the 4-week cruise, the sea ice extent varied as indicated by the contoured white areas:' and replace 'the white shadow' by a fourth contoured area.
3. Check all figures on their suitability for colour-blind people
4. P2, l33: replace 'of reducing anthropogenic CO2 in the atmosphere' with 'in regulating atmospheric carbon and acting as a net sink for anthropogenic carbon' or similar.
5. P2, l35: replace 'this status derives' by ' This uncertainty comes'
6. P2, l36: replace 'for' with 'because of'
7. P2, l38: move 'lying in the Indian Ocean section' to the next sentence and replace 'lying' by 'situated'
8. P2, l39-40: move 'With Cape Darnley …. to the east' to the end of the sentence or rephrase whole sentence, try to use the main verb as early as possible in a sentence
9. P2, l41: replace 'varies' by 'increases' (or does it go up and down?)
10. P3, l51-52: Add 'the': 'The Fram Bank and the Four Ladies Bank'
11. P3, l52: a spatial barrier for
12. P3, l54: replace 'part of it' by 'partly'
13. P3, l63-64: rephrase sentence to clarify the sequence of events
14. P3, l67: the importance for what? Replace 'carbon cycle' by 'carbon cycling'. This relates to comment 1 as well: how does studying the Prydz Bay relate to the SO carbon cycle?
15. P3, l69: use present tense where possible: 'is'
16. P3, l72: remove first word 'the'
17. P3, l77: Add 'A' before 'linear'. Clarify that it was not you doing this by adding 'In earlier studies, …'
18. P4, l78: What is a big scale? The entire Prydz Bay, the SO?
19. P4, l79: Start a new sentence at 'however'. Simplicity can be a good thing: why is calculating pCO2 based on SST and CHL insufficient? How do you know what controlling factors to select?
20. P4, l83: remove 'the' before 'February'

21. P4, l84: Is NN a type of neural network? The acronym NN is not used anywhere else in the manuscript – so not need to define it. What makes it artificial?
22. P4, l85: Remove 'been'
23. P4, l88: Add 'and' before 'chlorophyll'
24. P4, l92: Remove 'been' and replace 'a' before spatial-temporal by 'the'
25. P4, l97: Add the word 'cruise' after 'CHINARE'. Do the same on P4, l108.
26. P4, l98: replace 'to the early of March' with 'to early March'. Check general fluency of lines 97-99.
27. P4, l99: replace 'is show' by 'are shown'
28. P4, l101: here the authors suddenly discuss carbon absorption: the readers has not learned before that this area is considered to be a sink for carbon, so it would be could to introduce the reader to that earlier in the introduction
29. P4, l102: Replace 'followed' by 'follows'
30. P4, l104: Add ', and' and remove '.'
31. P4, l108: 'at the beginning of February 2015', did the cruise not extend into March? Why 'beginning'?
32. P5, l115: replace '$pCO_2$ in atmosphere' by 'atmospheric $pCO_2$'. Check also that each time you use the word $pCO_2$, that you use an italicised letter p (also in captions, and axes titles)
33. P5, l116/117: Replace 'in polar region' by 'in polar regions'
34. P5, l117: Move sentence 'Salinity records the physical processes' to later in the paragraph, because you first need to explain what salinity has to do with sea ice. It would also fit to explain to the reader why this is all relevant for a study of $pCO_2$.
35. P5, l117-118: Replace 'During freezing, salt is excluded … [] … brine rejection' with 'During freezing, brine is rejected from ice, thereby increasing sea surface salinity'.
36. P5, l119: replace 'to dilute' with 'thereby diluting'
37. P5, l125: Remove 'clearly'
38. P5, l127-128: 'the active biological process': Do you mean photosynthesis?
39. P5, l128-129: Explain the relationship between chlorophyll-a and biological productivity before you directly connect them and the consecutive effect on $pCO_2$ in this sentence.
40. P5, l129: Clarify that you used remote sensing data, and provide the reader with uncertainties associated with this method. Be consistent writing Modis either as Modis or MODIS.
41. P5, l130: Replace link by appropriate reference.
42. P5, l138-139: This sentence seems to repeat lines 121-122 on this page.
43. P5/6, l139-141: Rephrase sentence to make clear to the reader that there are two main methods in use, and what the advantages are of the 'difference criterion' method in the SO.
44. P6, l141: Add 'therefore' between 'we' and 'calculated'
45. P6, l142: Replace 'the' with 'on'
46. P6, l142-143: 'of with …' Do you mean 'of which'? I do not understand this sentence, sorry.
47. P6, l143-144: Why where the data gridded? They were point data from the CTD taken along the track, so why where they not already on the right spatial and temporal 'resolution' (do you mean interval?)?
48. P6, l150-151: Start with a capital letter t. Some words have disappeared from the caption.
49. P7, l161: Replace 'dimension' by 'dimensional'

50. P7, l163: 'Input variables', how do these relate to the boxes in Fig. 3? 'as a vector' is more fluent than 'in a vector form'
51. P8, l173: did not all your underway measurements include measurement of $pCO_2$?
52. P8, l178: Why did you quantify skewness and what did you do with the results? Is taking the logarithm an accepted method to improve the N coverage? Why does the coverage increase when taking the log?
53. P8, l186: Why is this not done for SST and SSS?
54. P9, l198: Add 'part of the' between 'second' and 'process'. Also, it is either each neuron or all neurons (i.e. is it plural or singular here?)
55.  P9, l213: What is meant with '8-d'? 8 dimensions, 8 days? If 8 days, why not 7 if used as weekly data?
56. P10, l243: Replace 'by two items' with 'using $\Delta pCO_2$ and the transfer velocity across the air-sea interface' or something similar.
57. P10, l246: Replace 'delta' with '$\Delta$'
58. P10, l247: What scaling factor are you talking about here? Is it in Eq. 2?
59. P10, l251: Check that equation has one format/font and denote units in []-brackets.
60. P10, l252: Check superscripts of $pCO_2\_air$ and $pCO_2\_sea$, also add 'and' before $pCO_2\_sea$ and end the sentence with 'respectively'
61. P10, l256: I am again confused by the use of the word regridding, you are working with sample data – why do you regrid? You mean you gridded the data from the point measurements you had of atmospheric $pCO_2$? What linear method did you use?
62. P10, l258-259: Do you mean you integrated the gridded flux over the area of Prydz Bay, taking into account the ice-free area only? How did you take ice into account?
63. P11, l267: No need to use the acronym AD if you only use it once
64. P12, l300: What is formed here? The subject of the sentence is the Shelf region, but a regions cannot be formed by modification of water.
65. P12, l305-306: If the region was ice-free, Fig. 5 cannot be correct?
66. P12, l314-315: When and where does the biological pump become the dominant factor setting the distribution of $pCO_2$? How do you know this is the main contributor to the $pCO_2$ variations?
67. P16, l371: What indicators did you use to conclude that the stability of the water was weak?
68. P16, l377: flew? Please rewrite this sentence.
69. P18, l395: $10^{12}$ gram=Tg
70. P18, l400: Pleas provide references to this statement and mention it earlier in the manuscript.
71. P18, l408-410: So does the region take up more carbon than on average in the ocean? I.e., is it a relatively large sink as compared to its area?

---

## Author Comment (AC1) · 17 Oct 2018

Summary:

Xu and colleagues investigate the regional sea surface $p\mathrm{CO_2}$ and air-sea flux in the Prydz Bay Antarctica using observations from the CHINARE cruise in February 2015. The authors divide the study regions into 3 sub-regions, based on the physical and biogeochemical controls of these sub-regions. Using a self-organizing map approach, the authors extrapolate the cruise data to the entire study region in order to estimate the carbon exchange of the Prydz Bay.

The Southern Ocean is still among the least observed and certainly least well understood ocean basins, hence I found this process study – investigating carbon variability and air-sea exchange in the Prydz Bay – to be very interesting and certainly relevant for the GB readership. More details on the strengths and weaknesses are listed below.

Strengths:

I found the manuscript and particularly the discussion of the processes comprehensive and logically built-up. The authors further make use of an appropriate and previously applied method based on machine learning (i.e. the SOM method) to extrapolate the cruise information to the full region of interest. They use independent validation data to test how well their approach reproduces observations from the SOCAT dataset and use this information to estimate the uncertainty of their integrated air-sea flux.

Weaknesses:

Up-front, I would like to note that there are several language issues – too many to be all named here (just one example: line 232: "In Pacific Ocean" should be "In the Pacific Ocean") – hence I do recommend English language editing.

During my review, I have encountered a few things that need clarification or some more information from the authors. They are listed from the most to least concerning. Additional comments (not of major concern) with line-numbers can be found at the end of this document:

1) Method section: At the moment, it is impossible for a reader who has not worked with the SOM approach to understand the methods section. Sentences like: "The SOM is trained using unsupervised learning to project the input space of training samples to a feature space (Kohonen, 1984), which is usually represented by grid points in tow-dimension space." Imagine a BG reader who is interested in the carbon exchange of the Prydz Bay but has never worked with a SOM. How is that person supposed to understand wording like"unsupervised, feature space, weight vector, training data, labeling data, etc." without reading several other papers first? As a SOM user I had no issues to follow this section but in my view, it has to be simplified for the more general BG audience. Furthermore, the authors miss to mention what distance function the SOM uses to detect the "winner neuron" (Euclidean distance maybe?). Furthermore, I don't think the phrase "resolve nonlinear relationships" (see abstract) is appropriate, since a SOM is a clustering algorithm that clusters based on similarities, but does not explicitly "resolve" a relationship.

**Response:** We have revised the introduction part about the SOM method to make it easy to understand what is SOM. And in the 2.2 section we have revised the sentences and adjusted the structure to make it easy for the reader to know how SOM works. In our SOM analysis we used Euclidean distance (the shortest distance) to select winner neurons and we have added this to the manuscript. We agree with the reviewer's suggestion and have changed the phrase 'resolve nonlinear relationships' to be 'to overcome a complex relationship among the biogeochemical and physical conditions in the Prydz Bay region'.

2) Training data: This links a bit to my point above but goes a bit more in-depth: I am not sure how data have been handled. On line 177 the authors state that the data have been "the four proxy parameters were logarithmically normalized" but table 1 suggests otherwise. In table 1 all values are absolute values. Besides that, I am not convinced that it makes sense to logarithmically normalize all 4 proxies. It makes sense for the skewed MLD and CHL-a but not really for salinity and temperature. Besides, I wonder how the normalization effects the distance function (which is not mentioned). Euclidean distances depend on the data-value range of each proxy. Also, what I am missing is a discussion why exactly the 4 proxies have been chosen? Why not sea surface height, wind speed, sea level pressure? What makes the 4 proxies so unique? I know they have been used by other authors, but the reader of THIS study needs this information.

**Response:** In table 1 all values are absolute values of the four proxies to show the value range. For the skewness and the N coverage percentage, the normalized data are shown in parenthesis. According to the change of skewness and N coverage percentage we found out only MLD and Chla data needed to be normalized for both the training and labeling dataset. Since we used Euclidean distance function to select the winner neuron and it depends on the data-value range of each proxy. The normalization for MLD and Chla dataset is to avoid weighting issue raised from the different magnitude among the variables.

In section 2.1 we have discussed the four proxies which will affect the distribution of $pCO_2$ in the surface sea water. The dissolution of $CO_2$ into water is mainly affected by temperature and pressure of water. The variation of salinity has little effect on the dissolution of $CO_2$. However the sea ice changed quickly in the study region and we chose salinity to be a proxy to simulate $pCO_2$. Moreover, in the region where local biology activities are active, $pCO_2$ will be affect strongly by photosynthesis. The mixed layer depth will prevent the upward mixing of nutrients and limits the biological production therefore we chose MLD as another proxy to simulate $pCO_2$. Sea surface height and sea level pressure are not major factors to the distribution of oceanic $pCO_2$. Wind speed is vital for the sea-air gas exchange and it is included in the air-sea flux equation.

3) Uncertainty: line389 states: "increased from week-1 (2.13 TgC) to week-2 (2.24TgC) due to increased wind speed." I was a bit disappointed here. First there is the effort to calculate uncertainties, then it is neglected in the text. Given the final uncertainty estimate, it is very unlikely that this regional difference of 0.1 TgC is significant. In general I suggest to add uncertainties wherever possible to avoid such misinterpretations.

**Response**: We have added uncertainties to the carbon uptake in section 3.4 and we have changed 'increased' to be 'changed mildly'.

4) Validation, comparison: I appreciate that the authors do a comparison with SOCAT data and include this in the overall flux uncertainty. I think that there need to be a bit more info in the text what cruise from SOCAT you are comparing to (this information is available on socat.info), or what the average spatial and temporal distance (which should be possible since a nearest grid method was used) between the cruises is. That certainly contributes to the mismatch as well. Otherwise, I was quite impressed by the relatively small (~22μatm)

difference. It might not sound small at first but your are comparing small special scale and high frequency temporal scale data based on the extrapolation of a single cruise. Therefore, 22μatm is impressive in my view. Furthermore, the RMSE tells the reader about the spread, but it would be valuable to add the mean (or absolute mean) difference between the SOM derived $CO_2$ and the SOCAT cruise. This would give you an indication of the bias.

**Response**: We have added the information of the cruise we selected from SOCAT in section 2.3. We have calculated the absolute mean difference between the SOM derived $CO_2$ and the SOCAT cruise. According to the validation, the SOM derived $pCO_2$ is generally lower than the SOCAT. Since the dataset from SOCAT does not cover the low- $pCO_2$ area towards the south, the precision might be of great uncertainty.

Methods section: On many occasions the authors re-grid data to the desired 0.1*0.1 resolution, but a bit more information on all data that were regridded and the algorithm would be appreciated. Ideally in form of a table. Additionally, I am missing the motivation why 0.1*0.1 was chosen. Why not 0.5*0.5 or even 0.05*0.05. Just to be clear, I don't suggest changing the resolution, but the text needs some motivation/technical explanation on why the current resolution was chosen that justifies all the data handling (i.e. regridding of proxy data)

**Response:** The 0.1*0.1 resolution of our study was desired according to the study area. It is a small area from 63E to 83E and 64S to 70S and the 0.1 resolution is the optimal. In the paper of Telszewshi et al. (2009), it was a basin-wide area from 9.5E to 75.5E and 10.5N to 75.5N, so their resolution was a 1 latitude by 1 longitude resolution. For a global area, Takahashi et al.(2012) chose 4*5 resolution. For our study area, it would be too rough if the resolution of 0.5, and the matrices would be too big if the resolution of 0.05.

The other data including remote sensing data and modeled data of different resolution were regridded to be the same resolution of 0.1 * 0.1 by Kriging method. We have added some explanation in the text. We think it is clear in the text.

Recommendation:

I have found this study to be interesting and to be of value to the BG readership. While I have raised some (partly major) concerns above I think that they can be resolved by the authors. I therefore recommend major revisions of the manuscript.

Specific and minor comments to the text:

1. Abstract line 14: Please also add the temporal resolution to the spatial resolution

   **Response:** We have added 'weekly' to the spatial resolution in abstract.

2. Abstract lines 27-29: This last sentence is out of context and is not something you can conclude from this study, hence it needs to be removed.

   **Response:** We have removed the last sentence.

3. Lines 32-33 reads "The role of the ocean south of 60S in the transport of $CO_2$ to or from the atmosphere is still uncertain despite of its importance of reducing anthropogenic $CO_2$ in the atmosphere" – that is a conflicting statement as it currently reads. If we know the importance of reducing atmospheric $CO_2$ how can its role be uncertain?

**Response**: It was a mistake. Here we mean 'the amount of carbon uptake in the ocean south of 60'. We have revised it.

4.  Lines 76-77: "Therefore, the direction of the sea-air $CO_2$ transfer is mainly regulated by the oceanic $pCO_2$" – this statement needs a reference
    **Response:** We have added the references needed.

5.  Line 84:"The SOM analysis, based on neural network (NN), a type of artificial neural network" – the second part (based on neural network) can be removed
    **Response:** It has been removed.

6.  Line 117: "Salinity records the physical processes" – When I read this sentence I also think of larger scale circulation and mixing in the context of physical processes, whereas this statement links to the follow-up discussion about brine rejection. Maybe a different term would be more appropriate.
    **Response:** It has been revised.

7.  Line 130: How was the interpolation done?
    **Response:** We gridded the chlorophyll-a data from Modis according the cruise track.

8.  Lines 133-136: "The mixed layer links the atmosphere to the deep ocean and plays a critical role in climate variability. Very few studies have emphasized the importance of accounting for the vertical mixing through the mixed layer depth" – Firstly, I disagree. Several studies have emphasized the importance of vertical mixing of carbon (but also nutrients, etc) through the mixed layer. Secondly, I caution the authors to mention the role in climate variability here. Their study does not resolve the necessary timescales to discuss either seasonal or interannual or decadal (whatever variability the authors refer to) variability.
    **Response:** We have made the correction and have removed the mention about the role in climate variability since in our study it didn't relate to that.

9.  Lines 154-155 'SOM based multiple non-linear regression' – This must have been a mistake or typo here, since the SOM (unlike e.g. a back propagation network) does not perform a regression (also not a non-linear one). Instead the SOM clusters data based on similar environmental conditions.
    **Response:** Yes, we agree the reviewer's suggestion and have removed 'multiple non-linear regression'.

10. Lines 194-195: "until the neural network sufficiently represents the nonlinear interdependence of proxy parameters used in training." – how is this judged? When do you know that its sufficient? I suppose this is judged by the number of SOM iterations, but how is set?
    **Response:** Because SOM analysis is a powerful technique to estimate $pCO_2$ from among the non-linear relationships of the parameters (Telszewski et al., 2009; ), actually, we presumed the nonlinear interdependence of proxy parameters are sufficiently represented after the

training procedure. Also, we used the som_make() function in the SOM toolbox for training data. Thus, we updated the sentence accordingly.

11. Line 215: "I could not figure out where the factor 30.8*10-4 comes from? Please explain in the text
    **Response:** The factor is induced according to the simplification of the equation. We have added the explanation in the text.

12. Line 264: "robustly divided" – I caution the authors here: How can you be sure the division is "robust"? Have you done any test that would proof robustness?
    **Response:** Three regions are divided according to the distribution of oceanic $pCO_2$. From the distribution of $pCO_2$ as shown in Fig.2-a there are three ranges. One is from 291.98 μatm to 379.31 μatm, the second is from 200 to 310μatm and the third is below 200μatm. We roughly divided the study region according to the three ranges of $pCO_2$ and the range of the depth of water in the Prydz Bay region. It was a mistake to use the word 'robustly'.

13. Lines 281-282: "region atmospheric $pCO_2$ was stable from 374.6μatm to 387.8μatm" That is a difference of 13μatm – I would not call this stable at all! I suppose this difference is largely the result of sea level pressure variability and relative humidity in the surface layer, hence it would be interesting to see the molar fractions (in ppm) for comparison if available.
    **Response:** We don't have sea level pressure data and relative humidity in the surface layer. We have revised this sentence and removed 'stable'.

14. Line 285: "biological consume" – should be "biological uptake"
    **Response:** It has been revised.

15. Line 318-319:"for a same period" – This would be important information. Furthermore, have you considered ARGO biogeochemistry floats from the SOCCOM array? They are deployed since 2013 and may add some additional independent estimate. This might however be beyond this manuscript.
    **Response:** Thanks for letting us know the SOCCOM. We have searched from SOCCOM but we can't find dataset useful for our study. However SOCCOM is a helpful website and we will turn to it when we other analyses in the Southern Ocean next time.

16. Figure 4b: It would be easier visible if x-axis and y
    **Response:** We have changed the x-axis and y to be the same range.

---

## Author Comment (AC2) · 17 Oct 2018

General comments

The manuscript 'variation of Summer Oceanic $p\text{CO}_2$ and Carbon Sink in the Prydz Bay Using SOM Analysis Approach' by Suqing Xu et al. presents their cruise data plus its analysis regarding oceanic and atmospheric $p\text{CO}_2$ and the related air-sea $p\text{CO}_2$ flux. The results can potentially be of interest to readers interested in the Southern Ocean carbon cycling, and its variability in time and space. It also provides an opportunity to the authors to show a practical example of the application of SOM in biogeochemistry. In order for the manuscript to be appreciated by the biogeochemical community, the authors should provide a better description of its relevance and importance for the greater Southern Ocean. S I am not an expert on SOM or neural networks, I cannot judge the methodology on that method in detail. I should however be able to understand what is presented in section 2.2. and I find this difficult at times. Several times mention is made of methods (like 'a linear method' or 'Linear regression extrapolation method') without further information on what is done: This makes reproducibility of the work without consulting the authors impossible. Besides that, I unfortunately often find the language to be confusing/imprecise, and therefore recommend professional English language checking before resubmitting. The language made it more difficult for me to judge the value of the manuscript, and I expect I can provide a more in-depth review after the language is improved. The manuscript would also improve if it were shortened as compared to the current version, as there is enough space to increase the information density in the manuscript in my opinion.

Specific comments

1. The introduction

   The introduction thoroughly describes the geographic setting of the Prydy Bay. I appreciate this, but it makes the introduction unbalanced as the questions 'why is this study of relevance' and 'what is new' are only covered by a few sentences. The authors describe the issue that the manuscript wants to address, namely the sparse spatiotemporal coverage of the Southern Ocean (SO) carbon cycle. They also tell the reader that they address the issue using the SOM approach. However, to what extent does research on the Prydz Bay support our understanding of the SO carbon cycle? On page 2, line 38-39 it is mentioned that the Prydz Bay is the third largest embayment in the Antarctic continent. No other reasons are given for the study of in specific this bay: What makes this bay (potentially) important for the SO carbon cycle even though it is small as compared to the total surface area of the SO? To what extent is this Bay representative for the SO as a whole (or just other parts of the SO),i.e. do the authors think their approach or data are useful for and representative of other areas in the SO? Why was the month February chosen to do the cruise?

   **Response:** The Prydz Bay region is the third largest embayment in the Antarctic continent and one of the source regions of Antarctic Bottom water (AABW) as well as the Weddell Sea and the Ross Sea (Jacobs and Georgi,1977; Yabuki et al., 2006). Studies have reported that Prydz Bay is a strong carbon sink in the austral summer (Gibsonab and Trullb, 1999; Gao et al., 2008; Roden et al., 2013). It is important to study the carbon cycle in the Prydz Bay. We have revised this part and added the information. The Prydz bay is part of the SO. SOM has been applied to simulate oceanic $p\text{CO}_2$ to overcome a complex relationship among the biogeochemical and physical conditions. We chose the beginning of February to early March because we had the in situ measurements during that time.

In the first sentence, it is mentioned that the SO is important for anthropogenic $CO_2$ uptake. The authors cannot distinguish between natural and anthropogenic carbon fluxes based on their measurements: Some sentences should be added to describe that the SO is a natural source of carbon to the atmosphere, but a sink for anthropogenic carbon – and that both are highly variable but creating a net sink for total carbon over the past decades. Here an argument could be made for their own study and cruise, which aims to reduce the spatiotemporal sparsity of the data and get a better understanding of the variability of the contemporary $pCO_2$ and its driving mechanisms. The authors call the Bay a sink at several instance (for example P3, L101 and P5, L125): Some numbers from previous studies should be given to support the statement that the Bay as a whole is a sink for carbon before presenting your own results.

**Response:** Sentences have been added to describe the SO on its role for carbon dioxide. About our study and cruise, we have added the argument. Recently studies have shown that there is a strong carbon sink in Prydz Bay especially in summer and we have added the references to support the statement.

In Figure1, an inset could be added to visualize the location of Fig.1 on the Antarctic continent.

**Response:** For Fig.1, we have added an inset to show the location of the Prydz Bay in the Antarctic continent.

P3,L64-66: How does a marine ecosystem interact with the physical environment to make it complicated to study $pCO_2$? Clarify your statement, as it currently is imprecise.

**Response:** We have revised this sentence. Here we mean due to the special physical environment and complicated ecosystem, it is difficult to study the spatiotemporal variation of $pCO_2$.

When describing the methods, clarify that in situ data from the cruise are combined with remotely sensed data to arrive at a gridded product.

**Response:** We have revised to clarify that in situ data from the cruise are combined with remotely sensed data.

2.1 In situ data

Here the authors present how they took their underway measurements and present them in Fig.2. The first time I read this section, I missed a good structure: The section starts with an explanation of the cruise and instruments used (until line 115). Then, the following paragraphs came to me as a bit of a surprise. One could help the reader find a better flow through the text by explaining that there are several processes/water characteristics that can influence the $pCO_2$ flux (which is the topic of this study). Then, the sea ice paragraph(lines 116-120), the information on the SSS and SST collection (lines 132-end of section) come more naturally. It is important to defend why specifically these proxies/data are used to do your study (create a gridded $pCO_2$ map). Don't forget to start the title with a capital letter i. It is unclear tome whether the results presented in Fig.2 are 4-week mean results or how they are calculated from the 4 cruise legs: Add more information to both the caption and the text.

**Response**: The results presented in Fig.2 are the data along the track cruise when R/V Xuelong sailed from east to west from the beginning of February to early March. It has been added in the caption and the text. We have added the information to explain some processes that can influence the $pCO_2$ distribution in the text.

2.2 SOM method and input variables

This section is generally hard to follow, maybe partly because I am not familiar with SOM. It should be improved so that also people new to SOM are able to understand and appreciate what you have done. Which 'environmental parameters' and which 'observational dataset's (Fig.3) are used? Lines 205-220 (or even up to 228) could be moved up in order to introduce the reader earlier to the datasets. Then the authors can explain what they are used for and how.

**Response**: Thanks for the suggestion. We have reconstructed this section and make it more clear about the 'environmental parameters' and 'observational datasets ' in the text. We have also revised the sentence about SOM method to make it easier to be understood.

2.3 Validation of SOM derived oceanic $pCO_2$

This section raises a lot of questions from my side. To what extent is SOCAT comparable to your data? Are the data both summer data? Why do you talk about assimilating several years together, but then only take 2015 from SOCAT (line 239)? Could you maybe compare your data to a model estimate of $pCO_2$ for this region? Lines 232-235: How is the equilibrium between atmospheric and surface ocean $pCO_2$, do you mean $pCO_2$-disequilibrium? Why do you describe this if you did not apply this method after all?

**Response**: We use dataset from SOCAT for the same period, which is February 2015. The dataset from SOCAT for validation as shown in Fig4-a. We prefer in situ measurements to model output to validate our results.We have removed line 232-238. Line 232-238 was a discussion and we think it didn't relate to the text.

2.4 Carbon uptake in the Prydz Bay

This section is quite clear to me: You have combined wind speed data and your $pCO_2$ measurements to arrive at a flux using Eq 2. However, you should clarify 1) where you used a 'scaling factor' (P10, L247-248) (in Eq. 2?), and 2) that that used your SOM-based $pCO_2$ product to calculate $pCO_2$ in Eq.2 (did you?). In addition, you write that the transfer velocity is a function of wind speed and temperature (Line 245) and then you write about a gas transfer rate (Line 248) (=transfer velocity?) which you apply a scaling factor to. I am left with the question which gas transfer rate or velocity you have used / how you calculated it.

**Response**: The original Eq.2 was a simplified equation considering the unit conversion factor. Now we have added the original sea-air $CO_2$ flux equation in the text and we have revised this part and added some information.

3.1 the distribution of underway measurements

Here you present your underway measurements for three areas. On what basis did you divide the Prydz Bay in these subregions? You write the division is 'robust' (P11, L264): Did you test what effect the choice of your division has on your results? It would be helpful to the reader if you added a plot figure with the subdivision of the Prydz Bay into its three regions. Add units to all

numbers (especially salinity lacks the psu unit throughout this section). I assume you are describing the results that are visualized in Fig 2 in this section: you should make reference to it if this is the case. Throughout the text of this section, you should be more precise on whether the values are regional means, 4-week means, and how you calculated this (refer to the methods).When you say decrease or increase (like P12, L291), it is not always clear to me whether it decreases/increases in time or space or whether the mean is lower or higher than in the neighboring sub-region. This causes for example confusion when SST's 'vary sharply' (L293) but 'decreased slightly' just the sentence above (L291). The readability of this section may improve by summarizing your main results in a table. A sentence should be added either here on the methods where the relationship between chlorophyll-a (as remotely observed) and biological productivity is stated.

**Response**: Three regions are divided according to the distribution of oceanic $pCO_2$ and depth of water. From the distribution of $pCO_2$ as shown in Fig.2-a and Table.2 there are three ranges. One is from about 300μatm to 380μatm, the second is from 200μatm to 350μatm and the third is below 250μatm. We roughly divided the study region according to the three ranges of $pCO_2$ and the range of the depth of water in the Prydz Bay region. It was a mistake to use the word 'robustly'. We have made the change to the text.

We have added units to all numbers. We have added the subdivision lines on Figures. 5.

We have added the reference to Fig 2 in this section.

Section 3.1 was about the in-situ measurements and the average values we discussed were regional mean. We have added the information in the text to avoid the confusion about the numbers. A table was added to the text summarizing our main results. A sentence has been added here about the relationship between chlorophyll-a and biological productivity.

3.2 Quality and maps of SOM-derived oceanic $pCO_2$

You compare your results to SOCAT and calculate the RMSE. Could you also provide the R2 of the best-fit line (red line in Fig. 4b)? You say your RMSE is consistent but not as good as most of the neuron methods. Do you mean it is on the high side of the accuracies previously reported, or why is it not as good? Could you calculate/estimate how many extra data points you would need to gain an improved precision of your SOM approach? You could probably comment on the limited amount of data that retrieving more data is not realistic with the resources and time available. SOCAT is not perfect either: A comment on its limited overlap with your study area would be appropriate here. It is surprising that the SOM estimate is generally higher than the SOCAT one, as SOCAT does not cover the low- $pCO_2$ area towards the south. Did you sample your SOM-derived $pCO_2$ dataset on the SOCAT locations, or did you compare all SOCAT in the area to all your data points in Fig. 4b? The first would probably be a fairer comparison and provide a better outcome as well. Fig.4a could be plotted in the same way as Fig.2 to make it easier for the reader to compare the spatial coverage.

**Response**: Our RMSE is on the high side of the accuracies previously reported and the correlation coefficient has been added in the text. There are two reasons accounting for the precision. One is the limited spatial coverage of the in situ measurements to be labeled in SOM method. Increasing the spatial coverage of the labeling data will help to increase the precision of SOM derived oceanic $pCO_2$. The other one is the dataset from SOCAT is not sufficient neither for space overlap nor for time overlap. The best way to get an improved precision of the SOM approach is to have a

full coverage measurement in the study area. In our study, we selected the SOM derived oceanic $pCO_2$ according to the location of the datasets from SOCAT for validation. As mentioned in the text, SOM derived $pCO_2$ is generally lower than the SOCAT one. We have plotted Fig.4a as Fig.2.

3.3 Spatial and temporal distributions of SOM-derived $pCO_2$
Here I expect the presentation of your main result: the $pCO_2$ maps of Figure 6. However, the text mostly describes the sea ice situation of the region: Why is this done here? Maybe a different title would be more appropriate? If sea ice is a main driving factor for $pCO_2$, this should be argued using the results. If the authors could add regional sub-division lines on the maps in Fig. 6, it might be easier to argue for the chosen sub-division (i.e. Shelf region, etc).
**Response**: We agreed with the reviewer and have revised this section. This section is mainly about the result of SOM derived $pCO_2$. We have presented the spatial and temporal distribution of SOM derived $pCO_2$. We have added regional sub-division lines on the maps.

3.4 Carbon uptake in Prydz Bay
This section is quite clear, although it would be good to clarify when mean values are reported, and whether they are regional means or temporal means, or both. From the figure on page 17 (which has no number?) it is hard to read the $pCO_2$ changes: one could either present it as a table, or adjust the y-axis range. Please make sure the figure is suitable for the color blind (and check this throughout the manuscript): Use for example different shapes for the three different lines in the upper graph, and add shapes in the lower one.
**Response**: We have changed the figure to be a table and we have made the revised in the text.

Supplementary information
The text at the start of the SI is already used in the main text, I do not see the need to provide it twice, and would recommend to remove it from the SI.
Technical corrections
I made an effort to pick out the most important language issues. However, as recommended in the general comments, I would strongly advise the authors to revise their language throughout the manuscript and to have it checked before resubmitting.
1.  Try to prevent the use of the word 'it' throughout the manuscript: replace by the actual subject of the sentence.
    **Response:** We have made the changes in the text.

2.  Caption of Fig.1: replace 'The circulations in the ' by 'The ocean circulation in the '. Replace sentence 'The weekly sea ice extents for our study periods were overlapped on the cruise.' By 'During the 4-week cruise, the sea ice extent varied as indicated by the contoured white areas:' and replace 'the white shadow' by a fourth contoured area.
    **Response:** It has been replaced.

3.  Check all figures on their suitability for color-blind people
    **Response**: We have checked all the figures.

4.  P2, L33: replace 'of reducing anthropogenic $CO_2$ in the atmosphere' with 'in regulating

atmospheric carbon and acting as a net sink for anthropogenic carbon' or similar.

**Response:** It has been replaced.

5. P2, L35: replace 'this status derives' by 'This uncertainty comes'

**Response:** It has been replaced.

6. P2,L36: replace 'for' with 'because of'

**Response:** It has been replaced.

7. P2, L38: move 'lying in the Indian Ocean section' to the next sentence and replace 'lying' by 'situated'

**Response:** It has been moved and replaced.

8. P2, L39-40: move 'With Cape Darnley … to the east' to the end of the sentence or rephrase whole sentence, try to use the main verb as early as possible in a sentence

Response: It has been moved and rephrased.

9. P2, L41: replace 'varies' by 'increases' (or does it go up and down?)

**Response:** It has been replaced.

10. P3, L51-52: Add 'the': 'The Fram Bank and the Four Ladies Bank'

**Response:** It has been added.

11. P3, L52: a spatial barrier for

Response: It has been revised.

12. P3, L54: replace 'part of it' by 'partly'

**Response:** It has been replaced.

13. P3, L63-64: rephrase sentence to clarify the sequence of events

**Response:** It has been rephrased.

14. P2,L67: the importance for what? Replace 'carbon cycle' by 'carbon cycling'. This relates to comment 1 as well: how does studying the Prydz Bay relate to the SO carbon cycle?

**Response:**We have added the importance of study carbon cycling in the Prydz Bay and added the information about the Prydz Bay related to the SO carbon cycle in the introduction section.

15. P3, L69: use present tense where possible: 'is'

**Response:** It has been replaced.

16. P3, L72: remove first word 'the'

**Response:** It has been removed.

17. P3,L77: Add 'A' before 'linear'. Clarify that it was not you doing this by adding 'In earlier studies, …'

**Response:** It has been revised.

18. P4, L78: What is a big scale? The entire Prydz Bay, the SO?
    **Response:** We have revised and made it clear to be 'that alinear regression extrapolation method has been applied to expand the cruise data to study the carbon cycle in the Southern Ocean'.

19. P4, L79: Start a new sentence at 'however'. Simplicity can be a good thing: why is calculating $p$CO$_2$ based on SST and CHL insufficient? How do you know what controlling factors to select?
    **Response**: There are two opposing processes primarily govern CO$_2$ chemistry in seawater: sinking of biological products from the photic zone to deep-ocean regimes (i.e., the biological pump), and upward transport by upwelling deep waters of CO$_2$ and nutrients formed by the decomposition of biological debris (i.e., the physical pump). It is not sufficient to simulate oceanic $p$CO$_2$ based on SST and CHL in previous studies, of which the RMSE tended to be high. From our previous researches and other studies we chose SST, CHL, MLD and SSS to be the controlling factors and we have added the information in the text.

20. P4, L83: remove 'the' before 'February'
    **Response:** It has been removed.

21. P4, L84: Is NN a type of neural network? The acronym NN is not used anywhere else in the manuscript – so not need to define it. What makes it artificial?
    **Response:** NN is an abbreviation for neural network. Here artificial means artificial intelligence.

22. P4, L85: Remove 'been'
    **Response:** It has been removed.

23. P4, L88: Add 'and' before 'chlorophyll'
    **Response:** It has been added.

24. P4,L92: Remove 'been' and replace 'a' before spatial-temporal by 'the'
    **Response:** It has been removed.

25. P4, L97: Add the word 'cruise' after 'CHINARE'. Do the same on P4, L108.
    **Response:** They have been revised.

26. P4, L98: replace 'to the early of March' with 'to early March'. Check general fluency of lines 97-99.
    **Response:** It has been replaced.

27. P4, L99: replace 'is show' by 'are shown'
    **Response:** It has been replaced.

28. P4, L101: here the authors suddenly discuss carbon absorption: the readers have not learned before that this area is considered to be a sink for carbon, so it would be could to introduce the reader to that earlier in the introduction
**Response:** It has been revised and we have added the information that the Prydz Bay is a carbon sink in the introduction.

29. P4,L102: Replace 'followed' by 'follows'
**Response:** It has been replaced.

30. P4, L104: Add ', and' and remove '.'
**Response:** It has been revised.

31. P4, L108: 'at the beginning of February 2015', did the cruise not extend into March? Why 'beginning'?
**Response:** It has been revised. The cruise was from the beginning of February to early March.

32. P5, L115: replace '$pCO_2$ in atmosphere' by 'atmospheric $pCO_2$'. Check also that each time you use the word $pCO_2$, that you use an italicized letter p (also in captions, and axes titles)
**Response:** It has been revised.

33. P5, L116/117: Replace 'in polar region' by 'in polar regions'
**Response:** It has been replaced.

34. P5, L117: Move sentence 'Salinity records the physical processes' to later in the paragraph, because you first need to explain what salinity has to do with sea ice. It would also fit to explain to the reader why this is all relevant for a study of $pCO_2$.
**Response:** It has been revised.

35. P5, L117-118: Replace 'During freezing, salt is excluded … [] … brine rejection' with 'During freezing, brine is rejected from ice, thereby increasing sea surface salinity'.
**Response:** It has been revised.

36. P5, L119: replace 'to dilute' with 'thereby diluting'
**Response:** It has been replaced.

37. P5, L125: Remove 'clearly'
**Response:** It has been removed.

38. P5, L127-128: 'the active biological process': Do you mean photosynthesis?
**Response:**Yes and we have added information about the relationship between chlorophyll-a and biological productivity in the text.

39. P5, L128-129: Explain the relationship between chlorophyll-a and biological productivity

before you directly connect them and the consecutive effect on $p\mathrm{CO_2}$ in this sentence.

Response:

40. P5, L129: Clarify that you used remote sensing data, and provide the reader with uncertainties associated with this method. Be consistent writing Modis either as Modis or MODIS.

**Response**: We have clarified that we used remote sensing data from MODIS. The uncertainty associated was mentioned in the last paragraph in section 2.2.

41. P5, L130: Replace link by appropriate reference.

**Response**: We prefer the link to show where the data comes from.

42. P5, L138-139: This sentence seems to repeat lines 121-122 on this page.

**Response**: It has been deleted.

43. P5/6, L139-141: Rephrase sentence to make clear to the reader that there are two main methods in use, and what the advantages are of the 'difference criterion' method in the SO.

**Response**: It has been rephrased.

44. P6, L141: Add 'therefore' between 'we' and 'calculated'

**Response**: It has been added.

45. P6, L142: Replace 'the' with 'on'

**Response**: It has been replaced.

46. P6, L142-143: 'of with …' Do you mean 'of which'? I do not understand this sentence, sorry.

**Response**: Yes, we mean 'of which'.

47. P6, L143-144: Why where the data gridded? They were point data from the CTD taken along the track, so why where they not already on the right spatial and temporal 'resolution' (do you mean interval?)?

**Response**: Yes, we gridded the point data from the CTD taken along the track in interval and we have revised the sentence.

48. P6, L150-151: Start with a capital letter t. Some words have disappeared from the caption.

**Response**: It has been revised.

49. P7, L161: Replace 'dimension' by 'dimensional'

**Response**: It has been replaced.

50. P7, L 163: 'Input variables', how do these relate to the boxes in Fig.3?'as a vector' is more fluent than 'in a vector form'

**Response**: The input variables related to the environmental parameters in Fig.3. We have made it clear the input variables and the environmental parameters. We have also changed to

be 'as a vector'.

51. P8, L173: did not all your underway measurements include measurement of $pCO_2$?
    **Response**: The underway measurements included measurement of $pCO_2$. Here we mean: for the training process, the input environmental parameters are those from satellite and model data of 0.1 resolution. However, the measurement of $pCO_2$ was along the cruise track and it has a spatiotemporal limitation compared to satellite data.

52. P8, L178: Why did you quantify skewness and what did you do with the results? Is taking the logarithm an accepted method to improve the N coverage? Why does the coverage increase when taking the log?

53. P8, L186: Why is this not done for SST and SSS?
    **Response to No.52&53**: In table 1 all values are absolute values of the four proxies to show the value range. For the skewness and the N coverage percentage, the normalized data are shown in parenthesis. According to the change of skewness and N coverage percentage we found out only MLD and Chla data needed to be normalized for both the training and labeling dataset. Since we used Euclidean distance function to select the winner neuron and it depends on the data-value range of each proxy. The normalization for MLD and Chla dataset is to avoid weighting issue raised from the different magnitude among the variables.

    In section 2.1 we have discussed the four proxies which will affect the distribution of $pCO_2$ in the surface sea water. The dissolution of $CO_2$ into water is mainly affected by temperature and pressure of water. The variation of salinity has little effect on the dissolution of $CO_2$. However the sea ice changed quickly in the study region and we chose salinity to be a proxy to simulate $pCO_2$. Moreover, in the region where local biology activities are active, $pCO_2$ will be affect strongly by photosynthesis. The mixed layer depth will prevent the upward mixing of nutrients and limits the biological production therefore we chose MLD as another proxy to simulate $pCO_2$. Sea surface height and sea level pressure are not major factors to the distribution of oceanic $pCO_2$. Wind speed is vital for the sea-air gas exchange and it is included in the air-sea flux equation.

54. P9, L198: Add 'part of the' between 'second' and 'process'. Also, it is either each neuron or all neurons (i.e. is it plural or singular here?)
    **Response**: It has been added and corrected to be 'neuron'.

55. P9,L213: What is meant with '8-d'? 8 dimensions, 8 days? If 8 days, why not 7 if used as weekly data?
    **Response**: '8-d' meant 8 days here. Our study period was from the beginning of February to March 4. When we used 8 days as weekly it was proper to cover the study period.

56. P10, L243: Replace 'by two items' with 'using $pCO_2$ and the transfer velocity across the air-sea interface' or something similar.
    **Response**: It has been replaced.

57. P10, L246: Replace 'delta' with '△'

**Response**: It has been replaced.

58. P10, L247: What scaling factor are you talking about here? Is it in Wq.2?
    **Response**: The scaling factor for the gas transfer rate is 0.251. It was not shown in Eq.2 because Eq.2 is a simplified equation taking into account the unit conversion factor. We have revised this part to make it clear.

59. P10, L251: Check that equation has one format/font and denote units in []-brackets.
    **Response**: It has been revised.

60. P10, L252: Check superscripts of $pCO_2$-air and $pCO_2$_sea, also add 'and' before $pCO_2$_sea and end the sentence with 'respectively'
    **Response**: It has been checked.

61. P10, L256: I am again confused by the use of the word regridding, your are working with sample data– why do you regrid? You mean you gridded the data from the point measurements you had of atmospheric $pCO_2$? What linear method did you use?
    **Response**: The atmospheric $pCO_2$ was of the cruise track. When we got the SOM derived oceanic pCO2 it was of 0.1*0.1 resolution. In order to calculate the air-sea flux we need to extrapolate the atmospheric $pCO_2$ to be the same 0.1*0.1 resolution. We used linear method.

62. P10, L258-259: Do you mean you integrated the gridded flux over the area of Prydz Bay, taking into account the ice-free area only? How did you take ice into account?
    **Response**: We have added the information to the text. The sea-air flux was calculated according to the proportion of ice-free area.

63. P11, L267: No need to use the acronym AD if you only use it once
    **Response**: It has been revised.

64. P12, L300: What is formed here? The subject of the sentence is the Shelf region, but a regions cannot be formed by modification of water.
    **Response**: It was a mistake and we have changed the subject to be 'water inside the Shelf region'.

65. P12, L305-306: If the region was ice-free, Fig.5 cannot be correct?
    **Response**: Fig.5 is correct and the ice shown in Fig.5 is permanent ice. We have revised the sentenced to be 'the most least ice-covered'.

66. P12, L314-315: When and where does the biological pump become the dominant factor setting the distribution of $pCO_2$? How do you know this is the main contributor to the $pCO_2$ variations?
    **Response**: The low oceanic $pCO_2$ was consistent with the high chlorophyll value in the Shelf region. For four weeks biological pump was the dominant factor setting the distribution of $pCO_2$. In the Shelf region other factors didn't show such pattern with oceanic $pCO_2$.

67. P16, L371: What indicators did you use to conclude that the stability of the water was weak?
   **Response**: The original sentence is not proper here. We have removed this sentence.

68. P16, L377: flew? Please rewrite this sentence.
   **Response**: It was a mistake. It should be 'flowing' and we have corrected it.

69. P18, L395: $10^{12}$gram=Tg
   **Response**: It has been revised.

70. P18, L400: Please provide references to this statement and mention it earlier in the manuscript.
   **Response**: The references have been added and we have added the information in the introduction.

71. P18, L408-410: So does the region take up more carbon than on average in the ocean? I.e., is it a relatively large sink as compared to its area?
   **Response**: Yes, this region takes up more carbon than on average in the ocean. Though small area, it is a relatively large sink. Taking into account the Prydz Bay is one of the resources of AABW (Antarctic Bottom Water), large amount uptake of atmospheric $CO_2$ may have an effect on the ocean acidification in the long run.

---

## Author Comment (AC3) · 22 Oct 2018

Many thanks to our respectable reviewer for his precious suggestions. We have made all the revise according to the suggestions and made some explanation. After the revise we found that our manuscript is of better quality. Please see the responses in detail in the attached file.

Please also note the supplement to this comment:
https://www.biogeosciences-discuss.net/bg-2018-276/bg-2018-276-AC3-supplement.pdf

**Fig. 1.**

[Figure]

**Fig. 2.**

[Figure]

Fig. 3.

---

## Author Comment (AC4) · 22 Oct 2018

Many thanks to our repectable reviewer. He or she has given us very precious suggestions and instructions to make our manuscript better. We agreed with the reviewer and have made all the revise according to the review comments. Our responses are in the attaced pdf file.

Please also note the supplement to this comment:
https://www.biogeosciences-discuss.net/bg-2018-276/bg-2018-276-AC4-supplement.pdf

[Figure]

Fig. 1.

[Figure]

Fig. 2.

[Figure]

Fig. 3.

---

## Author Response (AR2)

**Suggestions for revision or reasons for rejection (will be published if the paper is accepted for final publication)**

Dear authors,

1. In my initial assessment I raised 4 main points concerning methods and data handling, representation of uncertainty and validation. I am happy to say that you have well dressed most of the points raised. In particular, I appreciate that you now add uncertainties to all numbers (I suggest to even remove "mildly increased" based on the uncertainty). Furthermore data handling is much clearer now. Lastly, thank you for adding the cruise information from SOCAT.

   **Response:** We agree with the reviewer's suggestion and have removed 'mildly'.

   Overall, I believe the manuscript is technically sound and the results are well put in context. Therefore, I think the manuscript can be published after fixing a few remaining issues:

2. Methods: While the language is much simpler now and readability has improved, it still reads very difficult and it is certainly still the weakest bit. It is important for the reader to know why the inputs are in vector form. Between what are Euklidian distances calculated? What is a winning neuron? Maybe add a simple example with real data. Imagine and input vector (now with values I made up) with SST=3, SSS=32, CHl=1, MLD=10 and 2 neurons on the map with values (4,30,0.5,5) and (5,35,0,100) respectively. What is the winning neutron? after you have identified the winning neutron, what happens in step 1? and what happens in step 2? Then the link to the geography is also more visible, because the input vector values belong to a certain point in space, etc. That would be my suggestion here.

   **Response:** Because during the first step, each neuron's weight vectors, which are linearly initialized, are repeatedly trained by being presented with the input vectors. The inputs are prepared in vector from.   We have added more explanation in our manuscript. More detailed procedure comes from Nakaoka et al., 2013, we have added this reference in the text. We have added the equation to calculate the Euclidean distances to select a winner neuron as Eq.2 and added the explanation in the manuscript.

3. On line 142-144 I read: "The solubility of $CO_2$ is affected by temperature and salinity in the water as well as biological activities, such as phytoplankton taking up $CO_2$ through photosynthesis and organisms releasing $CO_2$ through respiration (Chen et al., 2011)." - I suppose this is a mistake from the authors (maybe awkward phrasing) as biology is not influencing the solubility.

   **Response:** We have rephrased this part.

4. Throughout the manuscript spaces are missing (see as example abstract line 15 and line 27 last word(s) of the line. This is reoccurring but nothing major.

   **Response:** It is a version problem. When our manuscript is opened in Word 2016 it is OK however when it is opened in other version the problem will occur. We have saved our manuscript to be in doc instead of docx and hope it will be OK.

5. lines 31-32: This is still awkwardly phrased as carbon uptake implies that the flux is from (and not "to or from") the atmosphere. Best is to remove the phrase "during the transport of $CO_2$ to or from the atmosphere" as a whole.
   **Response**: It has been removed.

6. line 88: satellites actually do measure $CO_2$ now - just not in the ocean (see GO-sat or OCO-2). So please say "measure sea surface $pCO_2$"
   **Response**: We have changed it.

7. line 139: remove of carbon cycling.
   **Response:** It has been removed.

8. line 169-170: please add a reference of some more text on the different methods. (better a reference)
   **Response:** A reference about different methods to calculate MLD has been added.

9. line 250: explain the Kriging method? or add a reference
   **Response:** The Kriging method we use is in SURFER software (version 7.3.0.35) and we have added it in the manuscript.

10. line 473: replace "could be" with "can be" - you just did.
    **Response:** It has been replaced.

[revised manuscript text omitted]